# Half-duplex and full-duplex interference mitigation in relays assisted heterogeneous network

**Moubachir Madani Fadoul** , **Chee-Onn Chow** *

Department of Electrical Engineering, Faculty of Engineering, University of Malaya, Kuala Lumpur, Malaysia

* cochow@um.edu.my

**Data Availability Statement:** Data are available from https://github.com/MoubachirMadani/MIMO_Half-Duplex_Full-Duplex_Relay.

**Funding:** This project was supported by the Ministry of Higher Education (MOHE) Malaysia

## Abstract

In a multicell environment, the half-duplex (HD) relaying is prone to inter-relay interference (IRI) and the full-duplex (FD) relaying is prone to relay residual-interference (RSI) and relay-to-destination interference (RDI) due to Next Generation Node B (gNB) traffic adaptation to different backhaul subframe configurations. IRI and RDI occur in the downlink when a relay is transmitting on its access link and interfering with the reception of a backhaul link of another victim relay. While the simultaneous transmission and reception of the FD relay creates the RSI. IRI, RDI, and RSI have detrimental effects on the system performance, leading to lower ergodic capacity and higher outage probability. Some previous contributions only briefly analysed the IRI, RSI, and RDI in a single cell scenario and some assumed that the backhaul and access subframes among the adjacent cells are perfectly aligned for different relays without counting for IRI, RSI and RDI. However, in practise the subframes are not perfectly aligned. In this paper, we eliminate the IRI, RSI, and RDI by using the hybrid zero-forcing and singular value decomposition (ZF-SVD) beamforming technique based on null-space projection. Furthermore, joint power allocation (joint PA) for the relays and destinations is performed to optimize the capacity. The ergodic capacity and outage probability comparisons of the proposed scheme with comparable baseline schemes corroborate the effectiveness of the proposed scheme.

## 1. Introduction

5G wireless network runs applications that require high demand for data rates. One of the solutions to solve the data rate requirement is to densify the network by deploying small cells. Such densification can reduce power consumption and offer higher spectral efficiency. Heterogeneous network offers significantly cost-effective means to enhance the capacity of the wireless cellular communication systems by permitting a variety of infrastructure nodes, including HD and FD relays, small cells, etc., to connect to the current multicell networks [1]. Small cells such as Femto and Pico varying sizes with low transmitted power are the most economic solutions. The main hurdle facing relay assisted heterogeneous network is the interference between nodes such as the relay may cause interference to another relay (IRI) and destination, referred to as relay-to-destination interference (RDI) in addition to relay residual-interference (RSI)

under the Fundamental Research Grant Scheme
(FRGS) (Grant number: FRGS/1/2020/TK0/UM/02/
4) and the funders had no role in study design,
data collection and analysis, decision to publish, or
preparation of the manuscript.

[2]. These HD and FD relay technology are suitable to be deployed in multicell network to provide coverage and diversity gain. The HD mode is dominating most current communication devices, where the HD relay employs time division or frequency division for backhaul and access link. Allocating different time slots for the transmission and reception, dissipates the worthy channel resources. On the other hand, by utilising FD relay and using the same communication resources, the wireless communication network can double the capacity if compared to the HD relay. Allowing the communicating devices to receive and transmit data simultaneously, using the same time slot through the same channel, brings many advantages. However, the main problem that affects the FD relay communication is the RSI which occurs between the receiving and transmitting antennas of the same node in addition to IRI and RDI.

## 1.1 Related work

Heterogeneous network is enabling a variety of infrastructures including a self-configurable relay, base-station etc., to connect to the current small cell networks in an unplanned manner. Increasing the ergodic capacity of wireless communication and efficient utilization of the scarce resources is a crucial claim for the next mobile network generation. Due to enormous growth in next mobile generation users, communication reliability becomes inevitable. The deployment of low-power nodes with irregular infrastructure is more demanding than the traditional cellular infrastructure with high power and regular structure [3, 4]. These small cells are suitable for deploying the HD and FD relay technology. Unfortunately, the HD mode is dominating most current communication devices, where the HD employs frequency division duplexing (FDD) or time division duplexing (TDD). Therefore, the worthy resources are dissipated. On the other hand, by utilizing FD and using the same communication resources, the wireless communication network can double the HD ergodic capacity. Allowing the communicating devices to transmit and receive data simultaneously, using the same time slot through the same channel, brings many advantages. FD, however, at the physical layer can double the spectral efficiency, which is measured by the number of bits that reliably communicated per second per Hertz. Further, in a contention network, FD permits implementing collision detection mechanisms while transmitting. However, the main problem that paralyzes the FD relay communication is the RSI which occurs between the transmitting and receiving antennas of the same node in addition to IRI and RDI due to network heterogeneity [5].

To get insight into the heterogeneous network interference, interference alignment is a promising technique that has been proposed to achieve high network capacity by increasing the usable channel resource units, i.e. degree-of–freedom (DoF). The main idea of interference alignment is to consolidate the interference into smaller dimensions of signal space at each receiver and use the remaining dimensions to transmit the desired signals [6]. Reference [4] modeled the hierarchical Aircomp designs with IRI from other cells. The major problem is that without inter-relay coordination, the characterization of the whole system is considered not for a specific cell. As cellular network splitted in different small cells, network providers may interest in knowing the performance of specific cells only.

The implementation of the FD relay in the past was infeasible, due to the increasing noise floor at the receiving antennas that may exceed the limited dynamic range of the analog-to-digital converter (ADC) [5]. The early research papers considered only single antenna nodes to evaluate the performance of a hybrid HD/FD relay network [7]; further, the RSI channel's gain is assumed to be a constant value, which works only for digital cancellation [8]. An infrastructure node network operates with two HD and FD is analyzed in [9, 10]. In FD wireless communication, a relay residual-interference cancellation is proposed for two models; where the first model assumed that the RSI is known precisely and the second model assumed that

the RSI is unknown [11]. With no bandwidth constraints and according to channel condition, the FD system can automatically switch its RSI cancellation. This system is restrained by practical limitations such as delay and signal attenuation [12]. An inband FD radio decreases the RSI level to noise floor is proposed in [13]. In an FD two-hop network, the relay acts as a multiple antennas FD node, the effect of resource allocation is studied [14], in which the impact of the relay distance from the transmitter, the number of antennas, and different RSI modes are investigated. In addition, maximizing the effective signal-to–interference-and-noise-ratio (SINR) leads to minimizing the outage probability, and the optimal choice between the HD and FD is calculated [10, 15]. In a small cell network and under certain power constraints, the spectral efficiency is maximized by a joint beamforming design [16–18]. The ergodic capacity comparison between the FD and HD relay is evaluated after modeling the FD relay with RSI, it is shown that the FD outperforms the HD relay [19] at low SNR [10] via numerical simulation. The capacity trade-off of HD and FD is analyzed for the total system in a two-hop amplify-and-forward (AF) relay. Allowing some SINR degradation with the FD mode is preferable to using two-time slots to eliminate RSI with the HD mode [20]. The FD hybrid BF technique is proposed in [1], where the sum rate capacity is improved by approximately doubling it due to the successful cancellation of the strong self-interference power.

On the other hand, an interest in multi-antenna technology has been witnessed in the past few years, which provides higher capacity and improves network coverage. Since the source, relay and destination equipped with multiple antennas, beamforming technique can be performed. The latest research considers the IRI, RSI and RDI to be perfectly mitigated [21] and others focus on designing beamforming techniques to cancel the interference in MIMO FD relay systems as shown in Table 1. A MIMO beamforming technique such as ZF is applied to suppress the interference or maximize the useful signal [1]. The performance of cognitive MIMO relay is analyzed by deploying selective zeroforcing beamforming and phase alignment. The ergodic capacity bound of FD relay for two sources has been investigated in [22], in addition, calculating the channel ergodic capacity, for example, the ergodic capacity of multicast channels is analyzed in [23].

In the spatial domain, the self-interference mitigation schemes such as beam selection, antenna selection, minimum-mean-squared-error (MMSE), and nullspace projection are proposed, where the relay is equipped with transmit and receive beamforming matrices. However, in the ideal case with perfect channel information, only nullspace projection can eliminate the residual interference [8, 18]. By maximizing the SINR, [24] suppresses the relay self-interference substantially with less impact on the desired signal. For FD wideband AF MIMO relays, SINR maximization based on RSI mitigation is proposed in [25, 26]. The DF capacity is maximized via joint optimization for the digital and analog transceiver is considered [17], to suppress the RSI, an additional adaptive technique was designed based on additional hardware. However, the system did not capture the network heterogeneity.

The study of [27, 28] evaluated the SNR of separate cells and concluded that a relay node can further improve the ergodic capacity by relay placement and ignoring the interference from neighboring cells. In a single cell network, [10] analyzed the effects of different RSI and RDI levels, the system performs worse at high SNR in terms of both outage probability and ergodic capacity. Reference [29] did not consider any channel model, but only fixed channel gain, in lieu, the SINR value is set to 10 dB for all hops. In this paper, we proposed a simplified and yet effective joint interference mitigation scheme for heterogeneous networks consisting of FD communication system and HD communication system called hybrid method.

To further improve the system performance, power efficiency is a vital design consideration. Although power allocation algorithms have been widely studied in multiple-input multiple-output (MIMO) systems [14, 30], the existing schemes cannot be directly applied to

**Table 1. Summary of RSI, IRI and RDI mitigation schemes for DF MIMO relay assisted HD and FD schemes.**

| Reference Article | Approaches | Interference | Performance Metric | Features | Limitations | Relaying Topology |
|---|---|---|---|---|---|---|
| [24] | Opportunistic mode selection for uplink and downlink | RSI | Spectral Efficiency | Transmit power adaptation to enhance the opportunistic mode switching between AF and DF. | Single Antenna. Switching between HD and FD requires tight synchronization and results in a overall system delay | Single Relay |
| [25] | Study the effect of the capacity and Rician factor in a Rician MIMO relay channel | RDI | Average Capacity | The message is encoded in strongest eigen-subchannels based on ordered eigen-values of Wishart matrices, while other signals are generated in remaining spaces. | Separate between DF and AF. Only HD relays are studied | Two Relays |
| [26] | Distributed Beamforming for MIMO FD Relaying Network | RSI, IRI and RDI | Average Capacity & Outage Probability | Beamforming and joint Power Allocation | Only single stream is considered for the analysis. The capacity and the interference analysis have not provided | Two Relays |
| [27, 28] | Transmit/receive beamforming are designed to maximise capacity. | RSI | Average Capacity & Outage Probability | To obtain the optimum performance, hybrid relaying modes is proposed to switch between HD and FD relaying. Twice the HD relay capacity was achieved. | The interference channel is restricted to a single stream. Only signle interference mitigation is considered | Single Relay |
| [29] | Obtain an up- per and lower bound that the sum rate lies in between with high probabil- ity for MIMO dimension and large number of users | RDI | Average Capacity | The BS mit- igates the interference by joint processing. The capacity for large network have been derived and compared single cell and multicell | The outage and the power allocation algo- rithm behavior is not explored | Multiple Relays |
| [30] | The diver- sity gain is obtained by designing trans- mit/receive beamforming | RSI | Average Capacity | Feedback- assisted MIMO syst- em ploying both transmit and re- ceive codebooks for transmit and receive diversity order | The digital domain and analog domain each requires separate circuit which increase the hardware. The capacity and outage plots were not given. | Single Relay |
| [31] | Two-cell two-hop MIMO interference channel with HD relays | RDI and IRI | Degrees of Freedom | Relaying pro- tocol employs the alternate transmission strategy and interference cancellation method. The proposed relay- ing protocol has an advantage in the feedback overhead compared to the aligned interference neutralization method | It requires three time slots to complete the two-hop trans- mission. IRI channel does not captured and is assumed to be perfectly decoded. Lim- ited to HD relays only | Two Relays |
| [32] | Transmit and receive BF to optimise overall performance for dual-hop AF FD MIMO relay system | RSI | Average Capacity | Joint source/relay precoding to achieve higher capacity compared to HD relaying under channel estimation errors | Each symbol- vector will be received twice in two con- secutive time slots, which restricts the FD performance. The transceiver needs to be redesigned to exploit the extra receive diversity and cancel interfer- ence. It only explored RSI | Single Relay |
| [33] | Optimal transceiver and relay processing algorithms for an FD AF MIMO two –way relaying system | RSI | Average Capacity | Accounts for RSI mitigation at each node and uses iterative technique to estimate the error accumulated over time | The system requires two time slots which does not explore the full FD capacity. Channel inver- sion may not perform well in a one way two hop network due to network requirement | Single Relay |
| [34] | Performance of coopera- tive multicell downlink communica- tion aided by polarization- multiplexing under limited feedback con- straints | RDI and IRI | Average Capacity | Polarized antennas in combination with joint preprocessing at the BSs and relays is regarded as an efficient technique for the cooperative multicell down- link system to deal with the space constraints | The joint pro- cessing suffered from feedback and backhaul delays. | Single Relay |

(*Continued*)

**Table 1.** (Continued)

| Reference Article | Approaches | Interference | Performance Metric | Features | Limitations | Relaying Topology |
|---|---|---|---|---|---|---|
| [35] | Outage probability minimized by beamforming optimization for FD DF multi-antenna relaying in the presence of CSI errors | RSI | Outage Probability | Optimal re-lay receive and transmit beamforming directions, based on the max-ratio combining/max-ratio transmission strategy | It only consid-ered the out-age probability. Only single in-terference mit-igation is cap-tured | Single Relay |
| [36] | Designing linear beam-formers in the MIMO multi-way relay channel with clustered full data exchange | RSI, IRI and RDI | Average Capac-ity and Degrees of Freedom | Transmit and receive BF enhance the capacity. The relay projects the received signal into a subspace orthogonal to the effective channels of other clusters. Analyzing various signal alignment patterns | The network studied as a cluster which will not give insights about each node. All nodes are subjected to a unity power | Multiple Relays |
| [37] | Joint source/relay precoding for dual-hop AF FD-MIMO relay system | RSI | Average Capacity | Transmit and receive beamforming to optimise overall performance with channel estimation error | Large codebook size increases complexity due to AF relay. Each symbol-vector received twice in two consecutive time slots, this restrict the FD performance. | Single Relay |
| Our Work | Joint transmit and receive BF to optimise overall per-formance for HD and FD MIMO relay heterogenous network | RSI, IRI and RDI | Average Capac-ity and Outage Probability | Transmit and receive BF enhance the ca-pacity. Hybrid zeroforcing and singular value decomposition (ZF-SVD) beamforming technique based on nullspace projection. Joint power allocation to optimize the capacity. | Several limita-tions of previ-ous works have been addressed | Multiple Relays |

multicell network because of RSI, IRI and RDI. This is due to the different power requirements from the backhaul and access links. Uniform power allocation at each node has been generally adopted for ease of analysis and computation Meanwhile, in MIMO systems without interference, it has been shown that waterfilling power allocation algorithm is optimal. However, in a MIMO relay network, individual power allocation and aggregate power allocation have been shown to further improve the system power efficiency in a relaying scheme [10, 14, 31] and can further be extended to multicell network.

## 1.2 Contribution

Motivated by the above mentioned limitations in Table 1 and owing to the practical HD and FD MIMO relaying network, the interference between nodes such as the relay may cause interference to other relay (IRI) and destination referred as relay-to-destination interference (RDI) in addition to residual self-interference (RSI) due to the use of multiple transmit and receive antennas in a limited space, and also due to network heterogeneity [2].

This paper considers a heterogeneous multicell network assisted by FD and HD relaying. Interference-aware transceiver beamforming (BF) matrices based on hybrid zeroforcing and singular value decomposition (ZF-SVD) beamforming technique at the relays and destinations are designed to jointly eliminate the RSI, IRI and RDI. The effectiveness of the proposed scheme has been investigated through Monte Carlo simulations. The results show that the proposed system can achieve better ergodic capacity, sum capacity, and outage probability performance than comparable baseline schemes. Due to the effectiveness of the proposed interference mitigation scheme, the proposed scheme achieves performance close to the ideal scheme without interference consideration. Finally, joint PA is proposed to further improve the system performance of our suboptimal scheme.

### 1.3 Paper outline

The rest of the paper is organized as follows. Next, we describe the system model in Section 2. The beamforming design is discussed in Section 3. The capacity of the proposed scheme is derived in Section 4. Comparable baseline schemes derived from the literature are given in Section 5. From Section 6 we further improve the performance of our proposed scheme by joint PA in Section 7. Numerical results are given in Section 8, and Section 9 offers the concluding remarks.

In this paper, vectors and matrices are respectively represented by boldface lowercase letters (e.g., $\mathbf{x}$) and boldface uppercase letters (e.g., $\mathbf{X}$). $\preceq$ and $\succeq$ denote the component wise inequality. "†" stands for the Moore-Penrose, $\{.\}^H$ represents conjugate transpose and $[x]^+ \triangleq max\{x, 0\}$, the expectation operator is given by $\mathbb{E}[.]$; $det(\cdot)$ stands for the determinant; $tr\{.\}$ is the trace of a matrix; the diagonal matrix that containing diagonal components $x_1, \cdots, x_m$ is denoted by $diag(x_1, \cdots, x_m)$; $\mathbf{X}_{N \times M}$ is the $N - by - M$ matrix with a $N$ rows and $M$ columns; $\mathbf{I}_{M \times M}$ denotes the identity matrix $M - by - M$.

## 2 System model

Consider a practical heterogeneous network scenario where the system suffers from RSI, IRI, and RDI, as depicted in Fig 1.

Such a network can be modeled in Fig 2, which shows the coexistence of FD and HD relay systems.

All HD sources, HD/FD relays, and low-mobility HD destinations which use half-duplex transmission instead of full-duplex to reduce the interference and share the channel by more than two nodes. (users) are equipped with multiple antennas. The source can't transmit to the destination directly, due to the effects of fading and shadowing, which makes sense for cases such as deploying the relay for coverage extension. In addition, the HD source $S_i$ wishes to communicate with HD destination $D_i$ through the FD relay $R_i$ in the cell $i$. While HD source $S_j$ wishes to communicate with HD destination $D_j$ through HD relay $R_j$ in the cell $j$ such that $S$

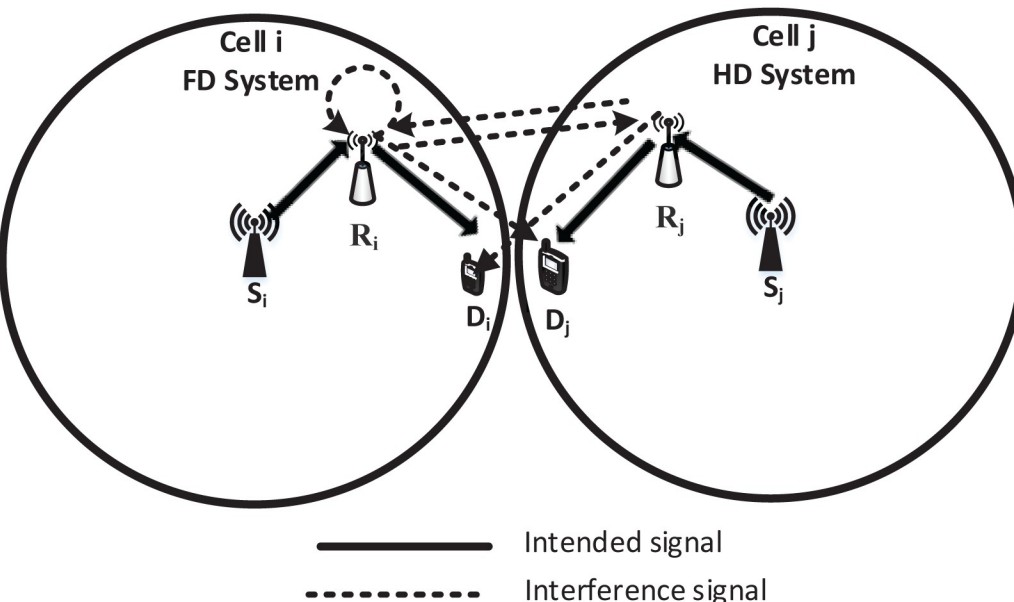

**Fig 1. Multicell network scenario shows the HD sources (S), HD/FD relays (R), and HD destinations (D).**

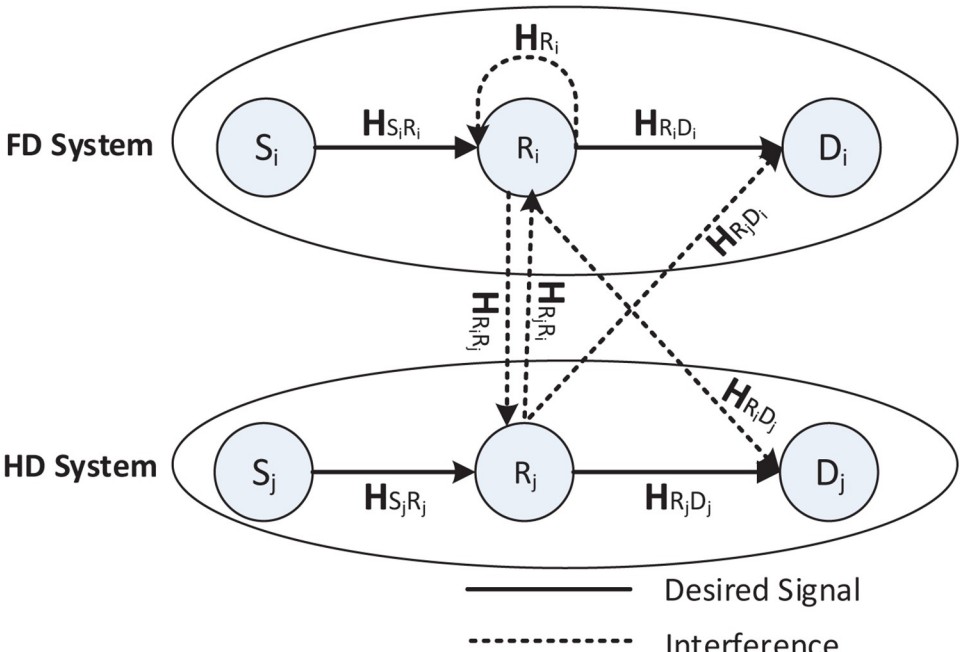

**Fig 2. Heterogeneous network model of two different systems; one is assisted by FD while the other is assisted by HD relay.**

won't interfere with $R$ and $D$, while the relays are deployed at the cell edge where the RSI, IRI and RDI courr. When $R_i$ is transmitting, it creates RSI to itself via the channel $\mathbf{H}_{R_i}$, IRI to the other relay $R_j$ via channel $\mathbf{H}_{R_iR_j}$ and RDI to the other node $D_j$ via the channel $\mathbf{H}_{R_iD_j}$. Likewise, when $R_j$ is transmitting, it creates IRI to the other relay $R_i$ via channel $\mathbf{H}_{R_jR_i}$ and RDI to the other node $D_i$ via the channel $\mathbf{H}_{R_jD_i}$ in addition to the receiver noise. Besides, additional interference may be caused by the Doppler effect due to for instance high-mobility planes, trains, etc., which could also be treated as part of the noise. All the channels are considered flat-fading spatially uncorrelated Rayleigh distributed. In other words, the entries of each channel matrix are independent and identically distributed (i.i.d.) complex Gaussian variables with zero-mean and unit variance. Further, the receiver channel state information (CSI) knowledge is assumed to be known.

The transmission protocol can be described in an odd and an even time slots. In the odd time slot $t$, the source $S_i$ transmits the message $\mathbf{x}_{S_i}$ to the FD relay $R_i$, and simultaneously the source $S_j$ transmits to the HD relay $R_j$. The FD relay $R_i$ simultaneously transmits and receives in the same frequency. This results in RSI through the channel $\mathbf{H}_{R_i}$ and IRI through $\mathbf{H}_{R_iR_j}$ to the other relay $R_j$. The following equations show the received message at FD relay $R_i$ and HD relay $R_j$, respectively

$$\mathbf{y}_{R_i}(\mathrm{t}) = \mathbf{H}_{S_iR_i}\mathbf{x}_{S_i} + \mathbf{H}_{R_i}\hat{\mathbf{x}}_{S_i} + \mathbf{z}_{R_i}, \tag{1}$$

$$\mathbf{y}_{R_j}(\mathrm{t}) = \mathbf{H}_{S_jR_j}\mathbf{x}_{S_j} + \mathbf{H}_{R_iR_j}\mathbf{x}_{S_i} + \mathbf{z}_{R_j}. \tag{2}$$

To cancel the interferences, the relay $R_i$ applies transmit BF while $R_j$ applies receive BF. The received signal at the relay $i$ and $j$ with the application of BF can be rewritten respectively as

$$\mathbf{y}_{R_i}(\mathrm{t}) = \underbrace{(\mathbf{H}_{S_iR_i}\mathbf{x}_{S_i} + \mathbf{z}_{R_i})}_{\text{desired signal plus noise}} + \underbrace{\mathbf{H}_{R_i}\mathbf{W}_{toi}\hat{\mathbf{x}}_{S_i}}_{\text{RSI}}, \tag{3}$$

$$\mathbf{W}_{roj}\mathbf{y}_{R_j}(\mathrm{t}) = \underbrace{\mathbf{W}_{roj}(\mathbf{H}_{S_jR_j}\mathbf{x}_{S_j} + \mathbf{z}_{R_j})}_{\text{desired signal plus noise}} + \underbrace{\mathbf{W}_{roj}\mathbf{H}_{R_iR_j}\mathbf{W}_{toi}\hat{\mathbf{x}}_{S_i}}_{\text{IRI}}. \tag{4}$$

Destination $D_i$ Within each time slot (which about usually hundreds mini-seconds), the geographic movement of low-mobility destinations (users) can be ignored. Hence, low-mobility destinations experienced zero Doppler spread. $D_i$ receives only from its desired relay $R_i$, because $R_j$ is kept silent during this period due to HD constraint. The design of the receive and transmit BF matrices will be discussed in the following Subsection. The received signal at $D_i$ is shown below

$$\mathbf{y}_{D_i}(\mathrm{t}) = \mathbf{H}_{R_iD_i}\hat{\mathbf{x}}_{D_i} + \mathbf{z}_{D_i}. \tag{5}$$

In the even time slot (t + 1), the source $S_i$ transmits the message $\mathbf{x}_{S_i}$ to the FD relay $R_i$. The FD relay $R_i$ simultaneously transmits and receives in the same frequency, this causes RSI through the channel $\mathbf{H}_{R_i}$. At the same time relay $R_i$ also receives IRI through channel $\mathbf{H}_{R_jR_i}$ from the other relay $R_j$. The following equation shows the received message at FD relay $R_i$,

$$\mathbf{y}_{R_i}(\mathrm{t}+1) = \mathbf{H}_{S_iR_i}\mathbf{x}_{S_i} + \mathbf{H}_{R_i}\hat{\mathbf{x}}_{S_i} + \mathbf{H}_{R_jR_i}\hat{\mathbf{x}}_{S_j} + \mathbf{z}_{R_i}. \tag{6}$$

To cancel the interferences, the $R_i$ and $R_j$ relays apply transmit BF. The received signal at the relay $R_i$ after applying the BF can be rewritten as

$$\mathbf{y}_{R_i}(\mathrm{t}+1) = \underbrace{(\mathbf{H}_{S_iR_i}\mathbf{x}_{S_i} + \mathbf{z}_{R_i})}_{\text{desired signal plus noise}} + \underbrace{\mathbf{H}_{R_i}\mathbf{W}_{tei}\hat{\mathbf{x}}_{S_i}}_{\text{RSI}} + \underbrace{\mathbf{H}_{R_jR_i}\mathbf{W}_{tej}\hat{\mathbf{x}}_{S_j}}_{\text{IRI}} \tag{7}$$

The received signal at destination $D_i$ and $D_j$ are

$$\mathbf{y}_{D_i}(\mathrm{t}+1) = \mathbf{H}_{R_iD_i}\mathbf{W}_{tei}\hat{\mathbf{x}}_{D_i} + \mathbf{H}_{R_jD_i}\mathbf{W}_{tej}\hat{\mathbf{x}}_{D_j} + \mathbf{z}_{D_i}, \tag{8}$$

$$\mathbf{y}_{D_j}(\mathrm{t}+1) = \mathbf{H}_{R_jD_j}\mathbf{W}_{tej}\hat{\mathbf{x}}_{D_j} + \mathbf{H}_{R_iD_j}\mathbf{W}_{tei}\hat{\mathbf{x}}_{D_i} + \mathbf{z}_{D_j}. \tag{9}$$

As shown in Eqs 8 and 9, the FD relay $R_i$ causes RDI to the $D_j$. The HD relay $R_j$ causes RDI to the other destination $D_i$ through the channel $\mathbf{H}_{R_jD_i}$. To eliminate the RDI, the destinations apply the receive BF matrices $\mathbf{W}_{redi}$ and $\mathbf{W}_{redj}$ as follows

$$\mathbf{W}_{redi}\mathbf{y}_{D_i}(\mathrm{t}+1) = \underbrace{(\mathbf{W}_{redi}\mathbf{H}_{R_iD_i}\mathbf{W}_{tei}\hat{\mathbf{x}}_{D_i} + \mathbf{W}_{redi}\mathbf{z}_{D_i})}_{\text{desired signal plus noise}} + \underbrace{\mathbf{W}_{redi}\mathbf{H}_{R_jD_i}\mathbf{W}_{tej}\hat{\mathbf{x}}_{D_j}}_{\text{RDI}}, \tag{10}$$

$$\mathbf{W}_{redj}\mathbf{y}_{D_j}(\mathrm{t}+1) = \underbrace{(\mathbf{W}_{redj}\mathbf{H}_{R_jD_j}\mathbf{W}_{tej}\hat{\mathbf{x}}_{D_j} + \mathbf{W}_{redj}\mathbf{z}_{D_j})}_{\text{desired signal plus noise}} + \underbrace{\mathbf{W}_{redj}\mathbf{H}_{R_iD_j}\mathbf{W}_{tei}\hat{\mathbf{x}}_{D_i}}_{\text{RDI}} \tag{11}$$

The definitions of symbol vectors and channel matrices are shown in the following.
$\mathbf{x}_S \in \mathbb{C}^{min(M_{Si}, M_{ri}) \times 1}$, and $\hat{\mathbf{x}}_S \in \mathbb{C}^{min(M_{Sj}, M_{rj}) \times 1}$ are the transmitted signals from the $S$ node with dimension $min(M_{Si}, M_{ri}) \times 1$ and $R$ node with dimension $min(M_{Sj}, M_{rj}) \times 1$, respectively.
$\mathbf{H}_{S_iR_i} \in \mathbb{C}^{N_{ri} \times M_{Si}}, \mathbf{H}_{S_jR_j} \in \mathbb{C}^{N_{rj} \times M_{Sj}}, \mathbf{H}_{R_i} \in \mathbb{C}^{N_{ri} \times M_{ri}}, \mathbf{H}_{R_j} \in \mathbb{C}^{N_{rj} \times M_{rj}}, \mathbf{H}_{R_iR_j} \in \mathbb{C}^{N_{rj} \times M_{ri}}, \mathbf{H}_{R_jR_i} \in \mathbb{C}^{N_{ri} \times M_{rj}},$

**Table 2. Summary of the protocol and the effective interferences.**

| Time Slots | Source (transmit) | Relay (receive) | Relay (transmit) | Destination (receive) | Interference |
|---|---|---|---|---|---|
| Odd time slot | $i, S_j$ | $R_i, R_j$ | $R_i$ | $D_i$ | RSI and IRI |
| Even time slot | $i$ | $R_i$ | $R_i, R_j$ | $D_i, D_j$ | RSI, IRI and RDI |

$\mathbf{H}_{R_j D_i} \in \mathbb{C}^{N_{di} \times M_{rj}}$, $\mathbf{H}_{R_i D_j} \in \mathbb{C}^{N_{dj} \times M_{ri}}$ and $\mathbf{H}_{R_i D_i} \in \mathbb{C}^{N_{di} \times M_{ri}}$, $\mathbf{H}_{R_j D_j} \in \mathbb{C}^{N_{dj} \times M_{rj}}$ are the channel gain matrices as shown in Fig 2. The summary of the odd and even time slots with their effective interferences are depicted in Table 2.

The power constraints on transmit signals are $\mathbb{E}[\mathbf{x}_S^\dagger \mathbf{x}_S] = 1$, $\mathbb{E}[\hat{\mathbf{x}}_R^\dagger \hat{\mathbf{x}}_R] = 1$. $\mathbf{y}_R \in \mathbb{C}^{N_r \times 1}$ and $\mathbf{y}_D \in \mathbb{C}^{N_d \times 1}$ are the received signals at $R$ and $D$ nodes. The $\mathbf{z}_R \in \mathbb{C}^{N_r \times 1}$ and $\mathbf{z}_D \in \mathbb{C}^{N_d \times 1}$ are independent circularly symmetric complex Gaussian noise vectors with distribution $\mathcal{CN}(0, N_0 \mathbf{I}_{Nr})$ and $\mathcal{CN}(0, N_0 \mathbf{I}_{Nd})$, and uncorrelated to $\mathbf{x}_S$ and $\mathbf{x}_R$. $\mathbf{I}_{Nd}$ and $\mathbf{I}_{Nr}$ are identity matrices of order $N_d$ and $N_r$ respectively. The transmit SNRs can be expressed as

$$\rho_{R_i} = \frac{P_{S_i R_i}}{N_0}, \rho_{D_i} = \frac{P_{R_i D_i}}{N_0}, \tag{12}$$

$$\rho_{R_j} = \frac{P_{S_j R_j}}{N_0}, \rho_{D_j} = \frac{P_{R_j D_j}}{N_0}. \tag{13}$$

## 3 Beamforming design

To mitigate RSI, IRI, and RDI, and to carefully ensure that the desired signal is not removed, the BF matrices have been designed. The BF matrices are projecting these interferences into the nullspace.

### 3.1 Relay transceiver beamforming design

To simultaneously mitigate the RSI and IRI at the odd time slot, the following transmit and receive beamforming matrices based on nullspace projection of the interference channels [3, 18] are proposed,

$$\mathbf{W}_{toi} = \text{Null}(\mathbf{H}_{R_i}) \tag{14}$$

$$\mathbf{W}_{roj} = \text{Null}(\mathbf{H}_{R_i R_j} \mathbf{W}_{toi}). \tag{15}$$

From Eqs 3 and 4, it is clear that in order to zeroforce the RSI and IRI, the transmit and receive BF matrix $\mathbf{W}_{toi}$ and $\mathbf{W}_{roj}$, respectively, are applied to project the RSI and IRI into the nullspace spanned by the interfering channels $(\mathbf{H}_{R_i}, \mathbf{H}_{R_i R_j} \mathbf{W}_{toi})$.

On the other hand, at an even time slot, the RSI and IRI are canceled by projecting the transmit BF matrices $\mathbf{W}_{tei}$ and $\mathbf{W}_{tej}$ to a nullspace of the channels $\mathbf{H}_{R_i}$ and $\mathbf{H}_{R_j R_i}$. Mathematically, the following relay transmit BF matrices are proposed

$$\mathbf{W}_{tei} = \text{Null}(\mathbf{H}_{R_i}), \tag{16}$$

$$\mathbf{W}_{tej} = \text{Null}(\mathbf{H}_{R_j R_i}). \tag{17}$$

In order to obtain non-zero nullspaces as in Eqs 14 to 17, the required dimensions are $M_{ri} \geq N_{ri} + min(M_{si}, N_{ri})$, $N_{rj} \geq min(M_{si}, N_{ri}) + min(M_{sj}, N_{rj})$ and $M_{rj} = N_{ri} + min(M_{sj}, N_{rj})$, and $N_{di} \geq min(M_{rj}, N_{dj}) + min(M_{ri}, N_{di})$, $N_{dj} \geq min(M_{ri}, N_{di}) + min(M_{rj}, N_{dj})$, where $min(M_s, N_r)$, and $min(M_r, N_d)$ are the number of the transmitted data streams which are also known as the rank of the channel. The dimension of the transmit BF matrices $\mathbf{W}_{toi} \in \mathbb{C}^{min(N_{ri}, M_{si}) \times M_{ri}}$, $\mathbf{W}_{tei} \in \mathbb{C}^{min(N_{ri}, M_{si}) \times M_{ri}}$, $\mathbf{W}_{tej} \in \mathbb{C}^{min(N_{rj}, M_{sj}) \times M_{rj}}$, and receive BF matrix $\mathbf{W}_{roj} \in \mathbb{C}^{M_{rj} \times min(N_{rj}, M_{sj})}$.

## 3.2 Destination beamforming design

At the even time slot, the receive BF matrices at the destination suppress the interference at the destination from unintended relay and maximize the desired signal. In other words, the BF matrices at the destination receivers $\mathbf{W}_{redi}$ and $\mathbf{W}_{redj}$ are designed to fulfill the ZF conditions: $\mathbf{W}_{redi} \mathbf{H}_{R_j D_i} \mathbf{W}_{tej} = \mathbf{0}$ and $\mathbf{W}_{redj} \mathbf{H}_{R_i D_j} \mathbf{W}_{tei} = \mathbf{0}$. The destination receives beamforming matrices that can be obtained as follows

$$\mathbf{W}_{redi} = \text{Null}\left(\mathbf{H}_{R_j D_i} \mathbf{W}_{tej}\right), \tag{18}$$

$$\mathbf{W}_{redj} = \text{Null}\left(\mathbf{H}_{R_i D_j} \mathbf{W}_{tei}\right). \tag{19}$$

At an odd time slot, no RDI occurs since there is only one transmitting relay. In this case, the transmit and receive BF matrices are designed for both relay $R_i$ and destination $D_i$ using conventional singular value decomposition (SVD). The SVD of the channel $\mathbf{H}_{R_i D_i}$ can be decomposed into three matrices $(\tilde{\mathbf{U}}_i \tilde{\mathbf{\Lambda}}_i \tilde{\mathbf{V}}_i^{\text{H}})$, where $\tilde{\mathbf{U}}_i$, $\tilde{\mathbf{V}}_i$ are unitary matrices, and $\tilde{\mathbf{\Lambda}}_i$ are diagonal matrix of $\mathbf{H}_{R_i D_i}$, sorted in descending order, whose diagonal elements $\tilde{\lambda}_1 \geq \tilde{\lambda}_2 ... \geq \tilde{\lambda}_{N_2}$, and the number of independent streams for SR-hop is $N_1 \leq min\{N_r, M_s\}$ and RD-hop $N_2 \leq min\{N_d, M_r\}$. The received signal at $R_i$ after SVD can be expressed as follows

$$\tilde{\mathbf{U}}_i^{\text{H}} \mathbf{y}_{R_i}(\text{t}) = \tilde{\mathbf{U}}_i^{\text{H}} \mathbf{H}_{S_i R_i} \tilde{\mathbf{V}}_i \mathbf{x}_{S_i} + \tilde{\mathbf{U}}_i^{\text{H}} \mathbf{z}_{R_i}, \tag{20}$$

$$\bar{\mathbf{y}}_{R_i}(\text{t}) = \tilde{\mathbf{\Lambda}}_i \mathbf{x}_{S_i} + \bar{\mathbf{z}}_{R_i}. \tag{21}$$

the dimension of the BF matrices $\mathbf{W}_{redi} \in \mathbb{C}^{N_{ri} \times (M_{ri} - min(N_{ri} - M_{si}))}$, $\mathbf{W}_{redj} \in \mathbb{C}^{N_{rj} \times (M_{rj} - min(N_{rj} - M_{sj}))}$.

## 4 Capacity performance

The system performance is measured using the total MIMO relay channel capacities according to Shannon formula. The total FD relaying capacity $C_{FD}$ is a sum of the FD capacity during the odd time slot $C_{FD}^o$ and the even time slot $C_{FD}^e$ whereas the HD relaying capacity $C_{HD}$ is defined by the minimum capacity for the odd time slot $C_{S_j R_j}^o$ and even time slot $C_{R_j D_j}^e$. However, these

capacities for the two hops are computed as follows

$$C_{HD} = \frac{1}{2}\min\left(C^o_{S_jR_j},\ C^e_{R_jD_j}\right), \tag{22}$$

$$C_{FD} = C^o_{FD} + C^e_{FD}, \tag{23}$$

$$C^o_{FD} = \frac{1}{2}\min\left(C^o_{S_iR_i},\ C^o_{R_iD_i}\right), \tag{24}$$

$$C^e_{FD} = \frac{1}{2}\min\left(C^e_{S_iR_i},\ C^e_{R_iD_i}\right). \tag{25}$$

## 4.1 Capacity of odd time slot—$C^o_{SR}$ and $C^o_{RD}$

With the instantaneous received SNR at the relay, the capacities of the odd time slot for the SR-hop $C^o_{S_iR_i}$ and $C^o_{S_jR_j}$ are affected by the RSI and IRI which are respectively given by

$$C^o_{S_iR_i} = \log_2\left|\mathbf{I}_{N_{ri}} + \frac{\rho_{R_i}}{M_{Si}}\left((\mathbf{H}_{S_iR_i}\mathbf{H}^H_{S_iR_i})\times((\mathbf{H}_{R_i}\mathbf{W}_{toi}\mathbf{W}^H_{toi}\mathbf{H}^H_{R_i})+\mathbf{I}_{N_{ri}}))^{-1}\right|. \tag{26}$$

$$C^o_{S_jR_j} = \log_2\left|\mathbf{I}_{N_{dj}} + \frac{\rho_{R_j}}{M_{Sj}}\left((\mathbf{W}_{roj}\mathbf{H}_{S_jR_j}\mathbf{H}^H_{S_jR_j}\mathbf{W}^H_{roj})\right.\right.$$
$$\left.\left.\times((\mathbf{W}_{roj}\mathbf{H}_{R_iR_j}\mathbf{W}_{toi}\mathbf{W}^H_{toi}\mathbf{H}^H_{R_iR_j}\mathbf{W}^H_{roj}+\mathbf{W}_{roj}\mathbf{W}^H_{roj}))\right)^{-1}\right|. \tag{27}$$

After applying the nullspace criteria Eqs 14 and 15, the Eqs 26 and 32 become

$$C^o_{S_iR_i} = \log_2\left|\mathbf{I}_{N_{ri}} + \frac{\rho_{R_i}}{M_{Si}}(\mathbf{H}_{S_iR_i}\mathbf{H}^H_{S_iR_i})\right|. \tag{28}$$

$$C^o_{S_jR_j} = \log_2\left|\mathbf{I}_{N_{rj}} + \frac{\rho_{R_j}}{M_{Sj}}(\mathbf{W}_{roj}\mathbf{H}_{S_jR_j}\mathbf{H}^H_{S_jR_j}\mathbf{W}^H_{roj})\right|. \tag{29}$$

while the capacity of the RD-hop $C^o_{R_iD_i}$ is given by

$$C^o_{R_iD_i} = \log_2\left|\mathbf{I}_{N_{di}} + \frac{\rho_{D_i}}{M_{ri}}\tilde{\mathbf{\Lambda}}^2_i\right|. \tag{30}$$

## 4.2 Capacity of even time slot—$C_{SR}^e$ and $C_{RD}^e$

With the instantaneous received SNR at the relay, the capacity of the even time slot for the SR-hop $C_{S_iR_i}^e$ is affected by the RSI and IRI, while $C_{S_jR_j}^e$ is free of interference. Specifically,

$$
\begin{aligned}
C_{S_iR_i}^e = \log_2 \Bigg| \mathbf{I}_{N_{ri}} + \frac{\rho_{R_i}}{M_{Si}}(\mathbf{H}_{S_iR_i}\mathbf{H}_{S_iR_i}^H) \\
\times((\mathbf{H}_{R_i}\mathbf{W}_{tei}\mathbf{W}_{tei}^H\mathbf{H}_{R_i}^H + \mathbf{H}_{R_jR_i}\mathbf{W}_{tej}\mathbf{W}_{tej}^H\mathbf{H}_{R_jR_i}^H) + \mathbf{I}_{N_{ri}}))^{-1} \Bigg|
\end{aligned}
\tag{31}
$$

$$
\begin{aligned}
C_{S_jR_j}^o = \log_2 \Bigg| \mathbf{I}_{N_{dj}} + \frac{\rho_{R_j}}{M_{Sj}}((\mathbf{W}_{roj}\mathbf{H}_{S_jR_j}\mathbf{H}_{S_jR_j}^H\mathbf{W}_{roj}^H) \\
\times((\mathbf{W}_{roj}\mathbf{H}_{R_iR_j}\mathbf{W}_{toi}\mathbf{W}_{toi}^H\mathbf{H}_{R_iR_j}^H\mathbf{W}_{roj}^H + \mathbf{W}_{roj}\mathbf{W}_{roj}^H)))^{-1} \Bigg|.
\end{aligned}
\tag{32}
$$

After applying the nullspace criteria Eqs 14 and 15, the Eqs 26 and 32 become

$$
C_{S_iR_i}^o = \log_2 \left| \mathbf{I}_{N_{ri}} + \frac{\rho_{R_i}}{M_{Si}}(\mathbf{H}_{S_iR_i}\mathbf{H}_{S_iR_i}^H) \right|.
\tag{33}
$$

$$
C_{S_jR_j}^o = \log_2 \left| \mathbf{I}_{N_{rj}} + \frac{\rho_{R_j}}{M_{Sj}}(\mathbf{W}_{roj}\mathbf{H}_{S_jR_j}\mathbf{H}_{S_jR_j}^H\mathbf{W}_{roj}^H) \right|.
\tag{34}
$$

while the capacity of the RD-hop $C_{R_iD_i}^o$ is given by

$$
C_{R_iD_i}^o = \log_2 \left| \mathbf{I}_{N_{di}} + \frac{\rho_{D_i}}{M_{ri}}\tilde{\mathbf{\Lambda}}_i^2 \right|.
\tag{35}
$$

## 4.3 Capacity of even time slot—$C_{SR}^e$ and $C_{RD}^e$

With the instantaneous received SNR at the relay, the capacity of the even time slot for the SR-hop $C_{S_iR_i}^e$ is affected by the RSI and IRI, while $C_{S_jR_j}^e$ is free of interference. Specifically,

$$
\begin{aligned}
C_{S_iR_i}^e = \log_2 \Bigg| \mathbf{I}_{N_{ri}} + \frac{\rho_{R_i}}{M_{Si}}(\mathbf{H}_{S_iR_i}\mathbf{H}_{S_iR_i}^H) \\
\times((\mathbf{H}_{R_i}\mathbf{W}_{tei}\mathbf{W}_{tei}^H\mathbf{H}_{R_i}^H + \mathbf{H}_{R_jR_i}\mathbf{W}_{tej}\mathbf{W}_{tej}^H\mathbf{H}_{R_jR_i}^H) + \mathbf{I}_{N_{ri}}))^{-1} \Bigg|
\end{aligned}
\tag{36}
$$

After applying the nullspace criteria in Eqs 16, 17 and 36 becomes

$$
C_{S_iR_i}^e = \log_2 \left| \mathbf{I}_{N_{ri}} + \frac{\rho_{R_i}}{M_{Si}}(\mathbf{H}_{S_iR_i}\mathbf{H}_{S_iR_i}^H) \right|.
\tag{37}
$$

Similarly, the capacity of destinations $C_{R_iD_i}^e$ and $C_{R_jD_j}^e$ with the RDI is respectively given as

$$
\begin{aligned}
C_{R_iD_i}^e = \log_2 \Bigg| \mathbf{I}_{N_{di}} + \frac{\rho_{D_i}}{M_{ri}}((\mathbf{W}_{redi}\mathbf{H}_{R_iD_i}\mathbf{W}_{tei}\mathbf{W}_{tei}^H\mathbf{H}_{R_iD_i}^H\mathbf{W}_{redi}^H) \\
\times((\mathbf{W}_{redi}\mathbf{H}_{R_jD_i}\mathbf{W}_{tej}\mathbf{W}_{tej}^H\mathbf{H}_{R_jD_i}^H\mathbf{W}_{redi}^H + \mathbf{W}_{redi}\mathbf{W}_{redi}^H)))^{-1} \Bigg|.
\end{aligned}
\tag{38}
$$

$$C_{R_j D_j}^{\mathrm{e}} = \log_2 \left| \mathbf{I}_{N_{dj}} + \frac{\rho_{\mathrm{D_j}}}{M_{rj}} ((\mathbf{W}_{redj} \mathbf{H}_{R_j D_j} \mathbf{W}_{tej} \mathbf{W}_{tej}^{\mathrm{H}} \mathbf{H}_{R_j D_j}^{H} \mathbf{W}_{redj}^{\mathrm{H}}) \right.$$
$$\left. \times ((\mathbf{W}_{redj} \mathbf{H}_{R_i D_j} \mathbf{W}_{tei} \mathbf{W}_{tei}^{\mathrm{H}} \mathbf{H}_{R_i D_i}^{H} \mathbf{W}_{redj}^{\mathrm{H}} + \mathbf{W}_{redj} \mathbf{W}_{redj}^{\mathrm{H}})))^{-1} \right|. \tag{39}$$

After applying the nullspace criteria in Eqs 18 and 19, Eqs 38 and 39 become

$$C_{R_i D_i}^{\mathrm{e}} = \log_2 \left| \mathbf{I}_{N_{di}} + \frac{\rho_{\mathrm{D_i}}}{M_{ri}} (\mathbf{W}_{redi} \mathbf{H}_{R_i D_i} \mathbf{W}_{tei} \mathbf{W}_{tei}^{\mathrm{H}} \mathbf{H}_{R_i D_i}^{\mathrm{H}} \mathbf{W}_{redi}^{\mathrm{H}}) \right|. \tag{40}$$

$$C_{R_j D_j}^{\mathrm{e}} = \log_2 \left| \mathbf{I}_{N_{dj}} + \frac{\rho_{\mathrm{D_j}}}{M_{rj}} (\mathbf{W}_{redj} \mathbf{H}_{R_j D_j} \mathbf{W}_{tej} \mathbf{W}_{tej}^{\mathrm{H}} \mathbf{H}_{R_j D_j}^{\mathrm{H}} \mathbf{W}_{redj}^{\mathrm{H}}) \right|. \tag{41}$$

## 5 Baseline schemes for comparison

In order to validate the performance of the proposed heterogeneous scheme, this Section derived the comparable baseline schemes from the literature for bench-marking.

### 5.1 Ideal FD relay scheme (no interference)

This scheme does not consider the effect of the interferences. This upper bounds the ideal multicell capacity [10, 18, 32–35]. The capacity of the ideal FD relaying scheme can be expressed as follows

$$C_{\mathrm{ID}} = \min(C_{SR,ID}, \ C_{RD,ID}), \tag{42}$$

$$C_{SR,ID} = \log_2 \left| \mathbf{I}_{N_{\mathrm{r}}} + \frac{\rho_{\mathrm{R}}}{M_S} \mathbf{H}_{SR} \mathbf{H}_{SR}^{\mathrm{H}} \right|, \tag{43}$$

$$C_{RD,ID} = \log_2 \left| \mathbf{I}_{Nd} + \frac{\rho_{\mathrm{D}}}{M_{\mathrm{r}}} \mathbf{H}_{RD} \mathbf{H}_{RD}^{\mathrm{H}} \right|. \tag{44}$$

### 5.2 FD relay without interference cancellation scheme (with interference)

This scheme considers the RSI, IRI and RDI effects without any suppression. Therefore, this scheme lower bounds the capacity of the proposed scheme [36]. The capacity of FD relay without interference cancellation can be expressed as

$$C_{FD_{IN}} = C_{FD_{IN}}^{\mathrm{o}} + C_{FD_{IN}}^{\mathrm{e}}, \tag{45}$$

$$C_{FD_{IN}}^{\mathrm{o}} = \frac{1}{2} \min \left( C_{S_i R_{i,IN}}^{\mathrm{o}}, \ C_{R_i D_{i,IN}}^{\mathrm{o}} \right). \tag{46}$$

$$C_{FD_{IN}}^{\mathrm{e}} = \frac{1}{2} \min \left( C_{S_i R_{i,IN}}^{\mathrm{e}}, \ C_{R_i D_{i,IN}}^{\mathrm{e}} \right). \tag{47}$$

**5.2.1 Capacity of odd time slot with interference—$C_{SR,IN}^{o}$.** With the instantaneous received SNR at the relay, the capacities of the odd time slot with interference for the SR hop $C_{S_iR_i,IN}^{o}$ and $C_{S_jR_j,IN}^{o}$ are affected by the RSI and IRI given by [32], Eq 22 as follows

$$C_{S_iR_{i,IN}}^{o} = \times\log_2\left|\mathbf{I}_{N_{ri}} + \frac{\rho_{R_i}}{M_{Si}}((\mathbf{H}_{S_iR_i}\mathbf{H}_{S_iR_i}^{H})((\mathbf{H}_{R_i}\mathbf{H}_{R_i}^{H}) + \mathbf{I}_{N_{ri}})))^{-1}\right|, \tag{48}$$

$$C_{S_jR_{j,IN}}^{o} = \times\log_2\left|\mathbf{I}_{N_{rj}} + \frac{\rho_{R_j}}{M_{rj}}((\mathbf{H}_{S_jR_j}\mathbf{H}_{S_jR_j}^{H}) \times ((\mathbf{H}_{R_jR_j}\mathbf{H}_{R_jR_j}^{H}) + \mathbf{I}_{N_{rj}})))^{-1}\right|. \tag{49}$$

**5.2.2 Capacity of even time slot with interference—$C_{SR_{IN}}^{e}$ and $C_{RD_{IN}}^{e}$.** With the instantaneous received SNR at the relay, the capacity of the even time slot with interference for the SR-hop $C_{S_iR_i,IN}^{e}$ is affected by the RSI and IRI [36], Eq 13 given by

$$C_{S_iR_i,IN}^{e} = +\log_2\left|\mathbf{I}_{N_{ri}} + \frac{\rho_{R_i}}{M_{Si}}(((\mathbf{H}_{S_iR_i}\mathbf{H}_{S_iR_i}^{H}) \times ((\mathbf{H}_{R_i}\mathbf{H}_{R_i}^{H} + \mathbf{H}_{R_jR_i}\mathbf{H}_{R_jR_i}^{H}) + \mathbf{I}_{N_{ri}})))^{-1}\right|. \tag{50}$$

Similarly the capacity of the destinations with interference $C_{R_iD_i,IN}^{e}$ and $C_{R_jD_j,IN}^{e}$ with the RDI [32, 34] is respectively given as

$$C_{R_iD_i,IN}^{e} = \times\log_2\left|\mathbf{I}_{N_{di}} + \frac{\rho_{D_i}}{M_{ri}}((\mathbf{H}_{R_iD_i}\mathbf{H}_{R_iD_i}^{H}\mathbf{H}_{R_jD_i}\mathbf{H}_{R_jD_i}^{H} + \mathbf{I}_{N_{dj}}))^{-1}\right|. \tag{51}$$

$$C_{R_jD_j,IN}^{e} = \times\log_2\left|\mathbf{I}_{N_{dj}} + \frac{\rho_{D_j}}{M_{rj}}((\mathbf{H}_{R_jD_j}\mathbf{H}_{R_jD_j}^{H}\mathbf{H}_{R_iD_j}\mathbf{H}_{R_iD_j}^{H} + \mathbf{I}_{N_{di}}))^{-1}\right|. \tag{52}$$

## 5.3 Ideal HD relay scheme (no interference)

This scheme does not consider the effect of the interferences. This upper bounds the ideal multicell capacity. The HD relay capacity of SR-hop and RD-hop is given by [32], Eq 17 and [33], Eq 8] as below

$$C_{HD} = \frac{1}{2}min(C_{SR,ID}, C_{RD,ID}). \tag{53}$$

## 6 Proposed scheme with waterfilling power allocation algorithm

In this Section, to maximize the total ergodic capacity, the effective transmit and receive BF matrices at the relay by using SVD have been designed. For the SR-hop, the SVD of the effective channels $\mathbf{H}_{SR}$ and $\mathbf{H}_{RD}$ can be decomposed into three matrices $(\mathbf{U}_i\mathbf{\Lambda}_i\mathbf{V}_i^{H}) = \text{SVD}(\mathbf{H}_{SR})$ and $(\tilde{\mathbf{U}}_i\tilde{\mathbf{\Lambda}}_i\tilde{\mathbf{V}}_i^{H}) = \text{SVD}(\mathbf{H}_{RD})$, respectively, as similar to [37], Eq 15 and [38], Eq 27. The SVD decomposes the channel into independent orthogonal sub-channels sorted in descending order, whose diagonal elements $\lambda_1 \geq \lambda_1... \geq \lambda_{N_1}$, and $\tilde{\lambda}_1 \geq \tilde{\lambda}_2... \geq \tilde{\lambda}_{N_2}$. The transmit signal $\hat{x}_S$ is multiplied by the right singular matrix $\mathbf{V}_i$ at the source. The received signal at the relay is multiplied by the left singular matrix $\mathbf{U}_i^{H}$.

From Eqs 33 and 34, the ergodic capacity of SR-hop can be expressed as

$$
C_{S_i R_i} = \log_2 \left| \mathbf{I}_{N_R} + \frac{P_{S_i R_i}}{N_0 M_S} \boldsymbol{\Lambda}_i \boldsymbol{\Lambda}_i^{\mathrm{H}} \right|.
\tag{54}
$$

The ergodic capacity of RD-hop from Eqs 40 and 41 can be rewritten as

$$
C_{R_i D_i} = \log_2 \left| \mathbf{I}_{Nd} + \frac{P_{R_i D_i}}{N_0 M_R} \tilde{\boldsymbol{\Lambda}}_i \tilde{\boldsymbol{\Lambda}}_i^{\mathrm{H}} \right|.
\tag{55}
$$

Thus, the MIMO relay channel is converted to non-interfering SISO sub-channels with non-equal power. Since the source and relay power is fixed, the power must be divided among these sub-channels. Waterfilling algorithm is shown to be the optimum power allocation algorithm [37], Eq 16. The ergodic capacity of SR-hop and RD-hop under the source transmit power $P_{SR}$ and relay transmit power $P_{RD}$ respectively with waterfilling can be shown as

$$
\bar{C}_{S_i R_i} = \sum_{i=1}^{N_1} \log_2 \left( 1 + \frac{P_{S_i R_i} \lambda_i}{N_0 M_s} \right),
\tag{56}
$$

$$
\bar{C}_{R_i D_i} = \sum_{i=1}^{N_2} \log_2 \left( 1 + \frac{P_{R_i D_i} \tilde{\lambda}_i}{N_0 M_r} \right),
\tag{57}
$$

where $\left( \mu_{RD} - \frac{N_0 M_r G G^H}{\lambda_2^2} \right)^+ = P_{SR_i}$ and $\left( \mu_{SR} - \frac{N_0 M_S}{\lambda_1^2} \right)^+ = P_{RD_i} \mu_{SR}$, $\mu_{RD}$ is determined to satisfy the power constraint $\sum_i P_{SR_i} = P_{SR}, \sum_i P_{RD_i} = P_{RD}$ respectively.

**Algorithm 1** Search Algorithm for solving Eq 58
```
1) Initialization: Set Δ = ζ(n) = 0.01, P_SR ⇐ P_t/2, P_RD = P_t − P_SR 2)
While|C_S_iR_i(P_SR) − C_R_iD_i(P_RD)| > ζ(n) do 3) if C_S_iR_i(P_SR) > C_R_iD_i(P_RD) then 4) P_SR ⇐ P_SR
− 0.01 5) P_RD ⇐ P_SR + 0.01 6) else 7) P_SR ⇐ P_SR + 0.01 8) P_RD ⇐ P_RD
− 0.018) The output is C_S_iR_i(P_SR) and C_R_iD_i(P_RD).
```

**Algorithm 2** Search Algorithm for solving Eq 58
```
1) Initialization: Set Δ = ζ(n) = 0.01, P_SR ⇐ P_t/2, P_RD = P_t − P_SR
2) While|C_S_iR_i(P_SR) − C_R_iD_i(P_RD)| > ζ(n) do
3) if C_S_iR_i(P_SR) > C_R_iD_i(P_RD) then
4) P_SR ⇐ P_SR − 0.01
5) P_RD ⇐ P_SR + 0.01
6) else
7) P_SR ⇐ P_SR + 0.01
8) P_RD ⇐ P_RD − 0.018)
The output is C_S_iR_i(P_SR) and C_R_iD_i(P_RD).
At the iteration number k, the term C_S_iR_i(k) and C_R_iD_i(k) are less or equal
the terms C_S_iR_i(k − 1) and C_R_iD_i(k − 1), respectively. As at each irritation
the step ζ(n) is subtracted from either P_SR or P_RD leading to the con-
vergence of the proposed algorithm.
```

## 7 Joint power allocation algorithm for source—Relay nodes

To further improve the performance, the total transmit power allocation at the source and relay can be optimized jointly based on the network power constraint. Maximizing the ergodic capacity of SR-hop and RD-hop is accomplished by formulating an optimization problem under a total network power constrain $P_t$.

Notice that Eqs 56 and 57, have been studied under the condition that $P_{SR}$ and $P_{RD}$ are fixed, i.e., the source and relay do not work cooperatively. This means that every node should

have their own power constraint which does not depend on the power consumption of other nodes. The advantage of this scheme is that the optimization problem can be calculated separately at each node. The disadvantage of separate power constraints at the source and relay is reducing the total capacity. Hence, a joint transmit power optimization of the source and relay would offer a higher capacity. Recall that the total system capacity is limited by the minimum of SR-hop and RD-hop. In such scenario, the transmit power at the weaker hop can be increased while the transmit power at the stronger hop is reduced. This motivates us to consider the joint PA for SR-hop and RD-hop to further improve the spectral efficiency, which simultaneously requires the solution of the following optimization problem

$$(P_{SR}^*, P_{RD}^*) = arg \max_{P_{SR}, P_{RD}} \bar{C} \tag{58}$$

$$s.t., P_{SR} + P_{RD} = P_t \tag{59}$$

The total system capacity is limited by the minimum SR-hop and RD-hop. To further optimize the system capacity, joint power allocation between the source and relay is considered. We further denote $\mathbf{A}_1 = \begin{bmatrix} p_1, & p_2, & ..., & p_{M_s} \end{bmatrix}^T$, in which $p_i = \begin{bmatrix} |v_{i,1}|^2, & |v_{i,2}|^2, & ..., \end{bmatrix}$ $|v_{i,M_s}|^2 \end{bmatrix}^T, \forall_i \in M_s$. Also $Also\mathbf{A}_2 = \begin{bmatrix} q_1, & q_2, & ..., & q_{M_r} \end{bmatrix}^T$, in which $q_i = \begin{bmatrix} |\tilde{v}_{i,1}|^2, & |\tilde{v}_{i,2}|^2, & ..., & | \end{bmatrix}$ $\tilde{v}_{i,M_r}|^2 \end{bmatrix}^T, \forall_i \in N_r.c = \begin{bmatrix} c_1, & c_2, & ..., & c_{M_s} \end{bmatrix}^T d = \begin{bmatrix} d_1, & d_2, & ..., & d_{M_r} \end{bmatrix}^T, b_1 = \begin{bmatrix} P_{SR,1}, & P_{SR,2}, & ..., & P_{SR,M_r} \end{bmatrix}^T$ and $b_2 = \begin{bmatrix} P_{RD,1}, & P_{RD,2}, & ..., & P_{RD,M_r} \end{bmatrix}^T$, the optimization problem can be reformulated as

$$(c^*, d^*) = arg \max_{c_i, d_i} \bar{C}, \quad such \ that \ A_1 c \preceq b_1, \ A_2 d \preceq b_2, \ F_1 c + F_2 d = h \tag{60}$$

where $c^*$ is the optimum value of $c$, and $d^*$ is the optimum value of $d$. From the dual composition, the partial Lagrangian [39], which can be obtained as

$$L(c, d, \gamma) = g_1^T c + g_2^T d + \gamma^T (F_1 c + F_2 d - h)$$
$$= (F_1^T \gamma + g_1)^T c + (F_2^T \gamma + g_2)^T d - \gamma^T h. \tag{61}$$

The dual function in Eq 58 is given by

$$q(r) = \inf_{c,d} \{L(c, d, \gamma) \mid A_1 c \preceq b_1, \ A_2 d \preceq b_2\}$$
$$= \inf_{A_1 c \preceq b_1} (F_1^T \gamma + g_1)^T c + \inf_{A_2 d \preceq b_2} (F_2^T \gamma + g_2)^T d - \gamma^T h. \tag{62}$$

The solution of Eq 59 can be obtained as

$$Max \ q(r)$$
$$Such \ that \gamma \succeq 0 \tag{63}$$

Eq 58 has a sub-gradient as follows

$$l(t) = -(F_1 c * \gamma(t)) - (F_2 d * \gamma(t)) + h, \tag{64}$$

where $\zeta(n)$ is the convergence step and $t$ is the iteration parameter. From Eq 63, the individual source and relay power is regulated by the master algorithm that is similar to the waterfilling power algorithm. To find the optimum value of $P_{SR}$ and $P_{RD}$, **Algorithm 1** is applied where $C_{SR}(P_{SR})$ is a function of the aggregate power of source and $C_{RD}(P_{RD})$ is a function of $P_{RD}$. To initiate the algorithm, we assume that $C_{S_i R_i}(P_{SR}) > C_{R_i D_i}(P_{RD})$.

## 8 Numerical results

To validate the proposed scheme with the comparable baseline schemes, Monte Carlo simulation results are provided and averaged over 10000 channel realizations. Equal power allocation (equal PA) assumes that the sources and relays have a unity transmit power and are subjected to an aggregate power constraint, i.e., $P_{SR} = P_{RD} = 1$, otherwise waterfilling and joint power allocation (Joint PA) splits the power between the source and relay. Recall that in the proposed scheme, the relay BF matrices are designed to mitigate the RSI, and IRI, while the destination BF matrices are designed to mitigate the RDI. Fig 3 shows the ergodic capacity for the proposed scheme with joint power allocation for $M_s = 2$, $N_{ri} = N_{rj} = N_r = 4$, $M_{ri} = M_{rj} = M_r = 6$ and $N_d = 4$ number of antennas compared to baseline schemes with equal power allocation. With the increase in SNR, the performance of the FD without interference cancellation remains flat because it is limited by the RSI, IRI, and RDI. The proposed scheme has well dealt with IRI, RSI, and RDI and the channel is tending to be well conditioned, due to joint PA [8, 18, 40, 41]. Moreover, a higher multiplexing gain (evident from the steep slope) is achieved by FD relaying, as compared to HD relaying. This is because the FD relay utilises the channel more efficiently. With proper RSI, IRI and RDI cancellation, the proposed FD relay based on nullspace projection achieves performance close to the ideal FD relay (no interference). The ideal FD relay capacity as shown in Section 5 is almost capacity achieving in scenarios where the $C_{SR}$ and $C_{RD}$ is sufficiently high because the RSI, IRI, and RDI are not considered. Even-though, the proposed scheme projects the RSI, IRI and RDI to the nullspace of the RSI, IRI and RDI

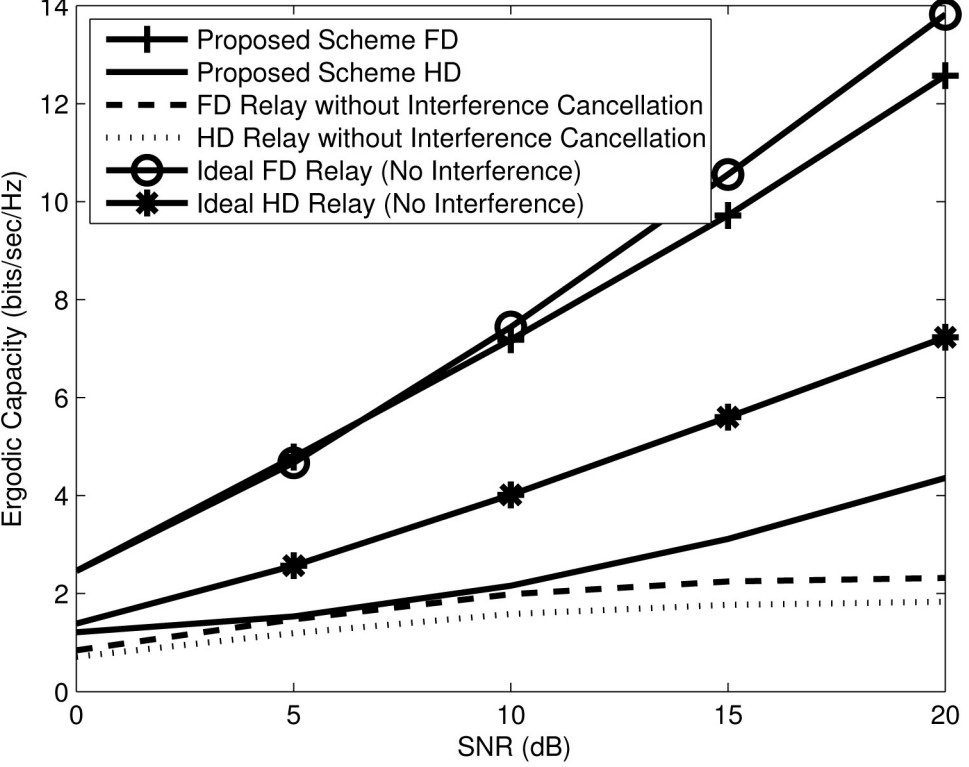

**Fig 3. Proposed scheme versus comparable baseline schemes.**

channels respectively, the proposed scheme achieves the same multiplexing gain as the ideal FD relay, evident from the slopes of the ergodic capacity curves.

Fig 4 shows the sum ergodic capacity by considering both the FD and HD relays-assisted networks. In general, the higher number of antennas produces higher capacity and multiplexing gain, because of high number of parallel streams that supported by the system. In fact, this is due to the increasing design freedom of the nullspace based relay transmit and receive beamforming design. A higher number of antennas is required at the relay and destination nodes if compared to the source node because the number of antennas must fulfill $N_{di} \geq min(M_{rj}, N_{dj})$ $+ min(M_{ri}, N_{di})$, and $N_{dj} \geq min(M_{ri}, N_{di}) + min(M_{rj}, N_{dj})$ to ensure the interference can be fully removed and provide sufficient degrees of freedom for the intended signal recovery [42, 43]. The slope of the curve denotes the diversity gain, which indicates how robust the system is when more antennas are added [44].

Fig 5 compares the sum capacity of HD and FD relays, bench-marked with the comparable baseline HD and FD schemes. We observe that the proposed scheme efficiently suppresses the interferences, and provides a significant capacity gain over the sum capacity of HD and FD system without interference cancellation for the whole range of SNR. The latter system is limited by the noise at low SNR and interference at high SNR [41, 43]. The sum capacity of the proposed scheme displays the same multiplexing gain (evident from the parallel slope) as the ideal interference free scheme [44].

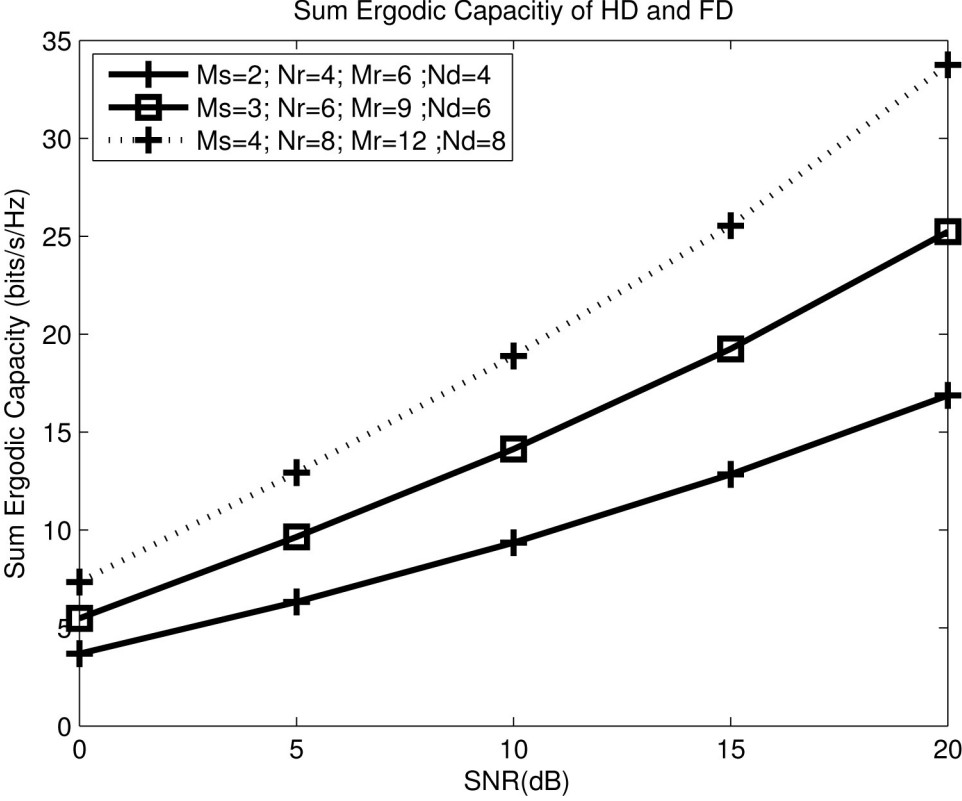

**Fig 4. The proposed sum capacity for FD and HD schemes with joint power allocation versus different number of antennas.**

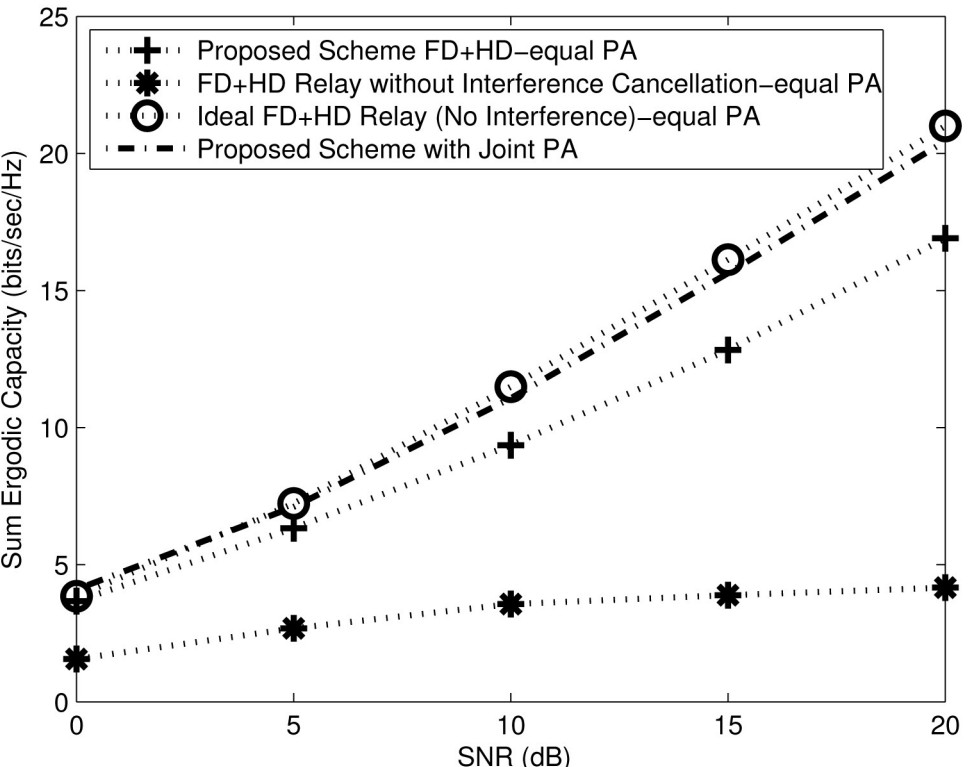

**Fig 5. Sum capacity for HD and FD vs SNR with $M_s = 2$, $N_r = 4$, $M_r = 6$ and $N_d = 4$ number of antennas.**

The outage probability for FD MIMO relaying is expressed as the probability that the instantaneous capacity falls below a given transmission rate threshold $\Re$ [33, 45]. Therefore, the outage probability is obtained by $min\left(P_{\text{out}}(\Re), \bar{P}_{out}(\Re)\right)$, mathematically expressed for equal PA and joint PA, respectively as

$$P_{\text{out}}(\Re) = P_r(C < \Re), \tag{65}$$

$$\bar{P}_{out}(\Re) = P_\text{r}(\bar{C} < \Re), \tag{66}$$

Fig 6 compares the outage probability of the proposed scheme with joint power allocation and comparable baseline schemes with equal power allocation for $M_s = 2$, $N_r = 4$, $M_r = 6$ and $N_d = 4$ antenna configurations and the target data rate $\Re$ is 3 bits/s. The outage performance of the comparable baseline FD and HD schemes with RSI, IRI, and RDI (without interference cancellation) experiences an outage floor at high SNR. In contrast, the outage probability of the proposed scheme decreases in proportional to the SNR, because the relay is able to cancel the RSI and IRI while the destination is able to cancel the RDI [42, 45]. From the slope of the curves, it can be seen that the proposed scheme achieves the same diversity order as the ideal scheme.

Fig 7 illustrates the outage probability vs different targeted data rates for fixed SNR. As it can be observed that the proposed scheme for HD and FD relaying with joint power allocation delivers performance closest to the ideal FD and HD Relay (no interference is considered). This means that the RSI, IRI and RDI have been well dealt with and the channels are tending

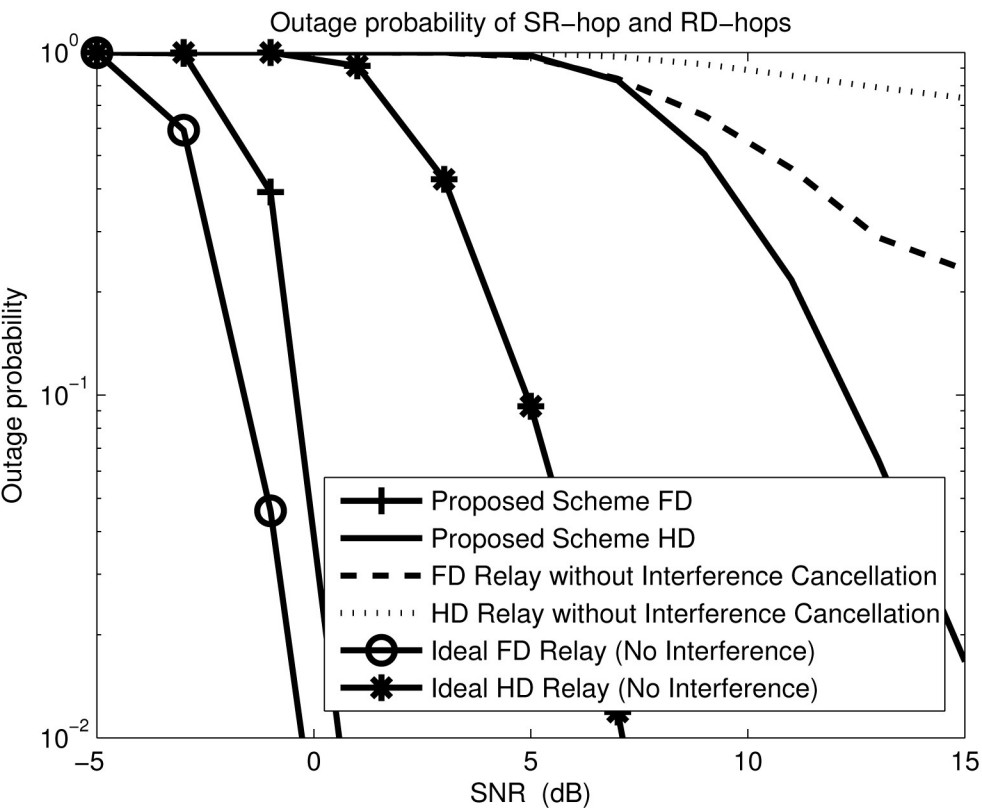

**Fig 6. Outage probability versus SNR for $M_s = 2$, $N_r = 4$, $M_r = 6$ and $N_d = 4$, $\Re = 3$ bits/sec.**

to be well conditioned due to joint power allocation. The proposed scheme FD after mitigating the interferences achieves close performance to the ideal FD relay. This performance could be seen in supporting high rates with lower outage probability as the ideal FD relay. The ideal FD or HD relay did not consider the interferences. On the other hand, the HD relay result is not surprising due to lower power transmission in multicell heterogeneous network. The proposed HD scheme shows much deviation as compared to the ideal HD relay. This poor performance is due to the HD constraints [33, 42, 45].

Denoting $d_1$ as the distance between the source and relay, $d_2$ as the distance between the relay and destination and $d_3$ as the distance between the relays, the average SNRs, which take into account the relay positions can be given below

$$\bar{\rho}_{R_i} = \frac{\rho_{R_i}}{d_i^{\alpha}}, \bar{\rho}_{D_i} = \frac{\rho_{D_i}}{d_i^{\alpha}}, \tag{67}$$

$$\bar{\rho}_{R_j} = \frac{\rho_{R_i}}{d_j^{\alpha}}, \bar{\rho}_{D_j} = \frac{\rho_{D_i}}{d_j^{\alpha}}, \tag{68}$$

where $\alpha$ is the path loss exponent.

Fig 8 shows outage probability vs SNR for $M_s = 2$, $N_r = 4$, $M_r = 6$ and $N_d = 4$ for fixed distances $d_1 = d_2 = 200m$ and varied distance $d_3$ to $200m$, $300m$ and $400m$. The path loss exponent, $\alpha$ is set to 4 and the target data rate at a fixed rate $\Re = 5$ bits/sec. It investigates the effect

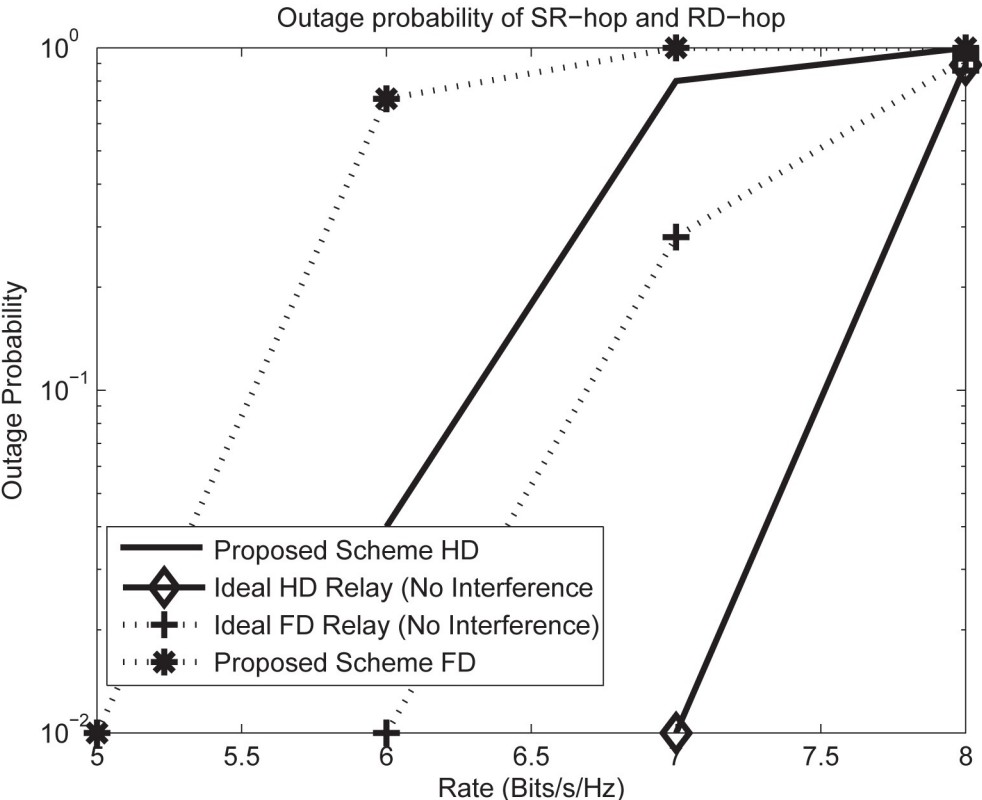

**Fig 7. Outage probability vs target data rate at fixed $SNR = 20dB$ with $M_s = 2$, $N_r = 4$, $M_r = 6$ and $N_d = 4$.**

of distance on the outage probability on the different relays. The probability of an outage for different distances decreases the channel gain resulting in a higher outage probability. There is a remarkable difference in the outage when $d_3 = 200m$ compared to $d_3 = 400m$. It's evident that the highest distance between the relays has the worst outage floor performance and requires more power to transmit. In contrast, when the distance between the relays decreases, the diversity order remains the same due to the efficient RSI, IRI and RDI mitigation. In particular, the beamforming is designed to project the received signals onto the nullspace of the interference channels [8, 40].

## 9 Conclusion

In this paper, a heterogeneous network assisted with MIMO HD and FD relays and affected by the interferences: RSI, IRI and RDI is investigated. An interference-aware BF scheme that simultaneously mitigates various combinations of RSI, IRI, and RDI is proposed. The detrimental effect of RSI, IRI, and RDI is removed using nullspace projection techniques at the transceivers. A heterogeneous network deployment becomes possible after canceling the RSI, IRI, and RDI, which offers a significant improvement over the sum capacity and outage probability of FD and HD schemes. This enables the FD to offer close to twice the conventional HD capacity. Further, from the slope of the sum rate, the proposed scheme achieves the same multiplexing gain as the ideal scheme. The results suggest that the ergodic capacity and outage probability can be significantly improved via joint power allocation with hybrid zeroforcing and singular value decomposition (ZF-SVD) beamforming technique.

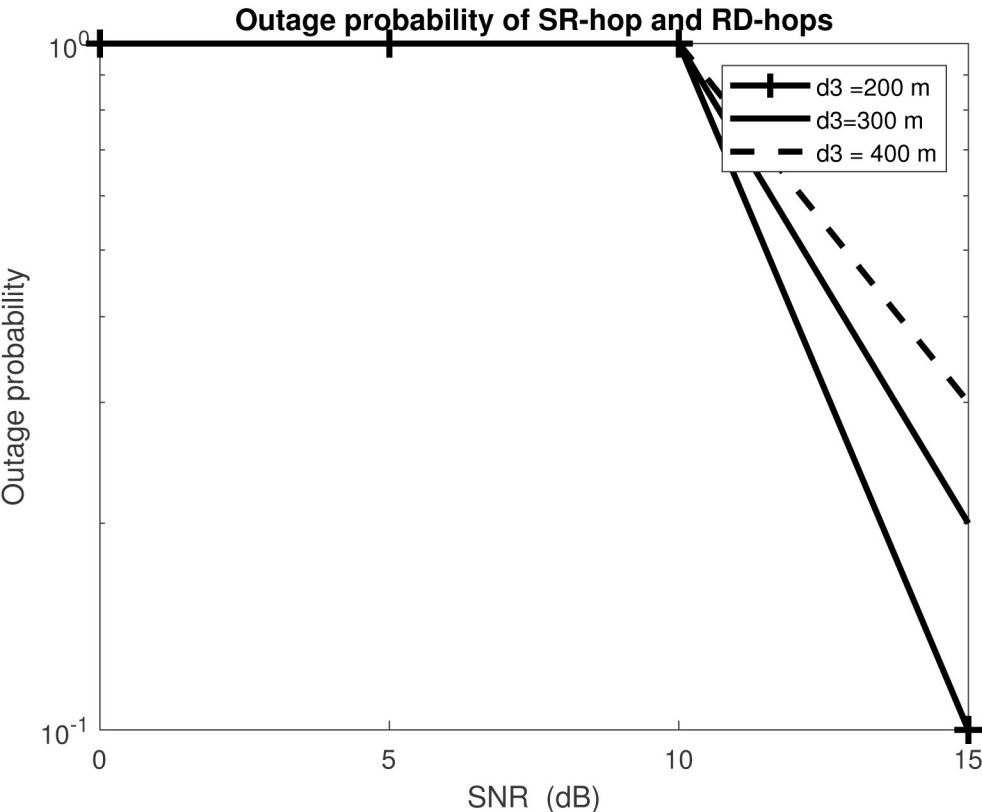

**Fig 8. Outage probability vs SNR with $M_s = 2$, $N_r = 4$, $M_r = 6$ and $N_d = 4$ for fixed distances $d_1 = d_2$ and varied distance $d_3$.**

## Acknowledgments

We would like to thank Assoc. Prof. Dr. Chee Yen Leow was Moubachir's Ph.D. supervisor from the School of Electrical Engineering, Faculty of Engineering, Universiti Teknologi Malaysia (UTM) for his insightful comments and suggestions.

## Author Contributions

**Conceptualization:** Moubachir Madani Fadoul, Chee-Onn Chow.

**Data curation:** Moubachir Madani Fadoul.

**Formal analysis:** Moubachir Madani Fadoul.

**Funding acquisition:** Moubachir Madani Fadoul, Chee-Onn Chow.

**Investigation:** Moubachir Madani Fadoul.

**Methodology:** Moubachir Madani Fadoul.

**Project administration:** Moubachir Madani Fadoul, Chee-Onn Chow.

**Resources:** Moubachir Madani Fadoul.

**Software:** Moubachir Madani Fadoul.

**Supervision:** Chee-Onn Chow.

**Validation:** Moubachir Madani Fadoul.

**Visualization:** Moubachir Madani Fadoul.

**Writing – original draft:** Moubachir Madani Fadoul.

**Writing – review & editing:** Moubachir Madani Fadoul.

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
