## [Decision Letter · Decision Letter 0]

16 Jan 2023

PONE-D-22-34295Half-Duplex and Full-Duplex Interference Mitigation in Relays Assisted Heterogeneous NetworkPLOS ONE

Dear Dr. Madani Fadoul,

Thank you for submitting your manuscript to PLOS ONE. After careful consideration, we feel that it has merit but does not fully meet PLOS ONE’s publication criteria as it currently stands. Therefore, we invite you to submit a revised version of the manuscript that addresses the points raised during the review process.

We look forward to receiving your revised manuscript.

Kind regards,

Praveen Kumar Donta, Ph.D.

Academic Editor

PLOS ONE

Journal Requirements:

Reviewers' comments:

Reviewer's Responses to Questions

**Comments to the Author**

1. Is the manuscript technically sound, and do the data support the conclusions?

Reviewer #1: Yes

Reviewer #2: Yes

2. Has the statistical analysis been performed appropriately and rigorously? 

Reviewer #1: Yes

Reviewer #2: No

3. Have the authors made all data underlying the findings in their manuscript fully available?

Reviewer #1: Yes

Reviewer #2: Yes

4. Is the manuscript presented in an intelligible fashion and written in standard English?

Reviewer #1: Yes

Reviewer #2: Yes

5. Review Comments to the Author

Reviewer #1: Authors eliminate the IRI, RSI, and RDI by using the hybrid zeroforcing and singular value decomposition (ZF-SVD) beamforming technique based on nullspace projection. Paper is technically sound and well written, i have few major comments:

1: It is important to show the effect on mobility.

2: Some important references are missing:

a:NOMA-Based Coordinated Direct and Relay Transmission With a Half-Duplex/ Full-Duplex Relay," in IEEE Transactions on Communications, vol. 68, no. 11, pp. 6750-6760, Nov. 2020,

2:"Heterogeneous Semi-Blind Interference Alignment in Finite-SNR Networks With Fairness Consideration," in IEEE Transactions on Wireless Communications, vol. 19, no. 4, pp. 2472-2488, April 2020.

Reviewer #2: This manuscript presents a framework for mitigation of the interference in a heterogenous networks where both full-duplex (FD) and half-duplex (HD) relays are exploited. The novel contribution of this work is the consideration of three kinds of interference (inter-relay interference, residual self-interference, and relay-to-destination interference in a multi-cell environment, especially in a heterogenous network. The authors also solve a joint power allocation optimization problem for the relays and the sources.

In general, the technical contribution of this work is good and the analysis is presented in a good organization. However, I do have some comments to improve the quality of this manuscript:

1. Although the problem considered in this work is novel and comprehensive, the solutions to the considered problem, such as beamforming, zero-forcing, and singular value decomposition, have been well-studied in the previous works. The beamforming designs for the relays and the destinations are quite independent on each other. Therefore, they are not so difficult to derived.

2. In the system model, the authors consider two adjacent cells, one of which uses half-duplex relay while the other uses full-duplex relay. This is not a typical situation. Why don't they consider two adjacent cells, both of which employ full-duplex relay?

3. Regarding the joint power allocation for source and relay nodes, why is there the total power constraint? These are two independence devices, so each should has its own maximum available power.

4. The baseline schemes for comparison are the HD and FD schemes without interference or with interference but without interference cancellation. These comparisons are trivial. Please compare the proposed method with related works that the authors have introduced.

5. In the Introduction section (page 3, lines 78 - 79), the authors claim that "by changing the level of self-interference, the FD ergodic capacity becomes similar to HD ergodic capacity". But then in the next sentence, they stated that "the ergodic capacity of FD achieves better performance than HD ergodic capacity". Please justify these sentences.

6. In the comments on Figure 3, the authors claim that "With the increase in SNR, the performance of FD without interference cancellation decreases". This is not compatible with the results in Figure 3.

7. In page 11, line 192, what does C^{(M_{S_j}, M_{r_j}) x 1} means?

8. Page 8, line 148: "(x)^{+} denotes x or 0"  this is not correct mathemmatically.

9. There are many typos and grammar errors in the manuscript:

- There are some duplication between the Introduction and the section on Related work, for example: "The main hurdle facing relay assisted ..." or "Small cells such as Femto and Pico ..."

- Page 3, lines 84-86: "... relay systems Table 1".

- Page 3, lines 101-102: "Though it achieves ... was not considered"  grammar error.

- The first paragraph of the subsection "Summary of Contribution" is not clear in meaning. Also, there is a typo: "Table ??".

- "moore-penrose pseudoinverse"  should be "Moore-Penrose ...".

10. Table 1 is too long, but the readers still cannot see what is the difference between the proposed work and those works in Table 1.

6. PLOS authors have the option to publish the peer review history of their article (what does this mean?). If published, this will include your full peer review and any attached files.

Reviewer #1: No

Reviewer #2: No

---

## [Author Response · Author response to Decision Letter 0]

4 Mar 2023

Manuscript ID PONE-D-22-34295

Firstly, we would like to thank the editor and the reviewers for their efforts in reviewing our manuscript. The comments are useful and constructive. Following the comments, we have made changes to improve the final manuscript. We also respond to individual concerns of the reviewers, where the comments are printed in italic, followed by the responses. In the revised manuscript, all changes are indicated by red colored fonts. The manuscript is edited in order to comply with length policy imposed by PLOS ONE Journal.

Response to Editor's Comments

We have ensured that the edited manuscript version meets PLOS ONE's style requirements, including those for file naming. The paper-generated code will be made available without restrictions upon publication of the work.

Reviewers' comments:

1. Is the manuscript technically sound, and do the data support the conclusions?

Reviewer #1: Yes

Reviewer #2: Yes

2. Has the statistical analysis been performed appropriately and rigorously?

Reviewer #1: Yes

Reviewer #2: No

3. Have the authors made all data underlying the findings in their manuscript fully available?

Reviewer #1: Yes

Reviewer #2: Yes

4. Is the manuscript presented in an intelligible fashion and written in standard English?

Reviewer #1: Yes

Reviewer #2: Yes

Response to Reviewers' Comments and Questions

Reviewer #1: Authors eliminate the IRI, RSI, and RDI by using the hybrid zeroforcing and singular value decomposition (ZF-SVD) beamforming technique based on nullspace projection. Paper is technically sound and well written, i have few major comments: 

1: It is important to show the effect on mobility. 

Response: Thank you for the suggestion, the relay mobility have been added with an additional figure (8).

2: Some important references are missing: a: NOMA-Based Coordinated Direct and Relay Transmission With a Half-Duplex/ Full-Duplex Relay," in IEEE Transactions on Communications, vol. 68, no. 11, pp. 6750-6760, Nov. 2020, 2:"Heterogeneous Semi-Blind Interference Alignment in Finite-SNR Networks With Fairness Consideration," in IEEE Transactions on Wireless Communications, vol. 19, no. 4, pp. 2472-2488, April 2020.

Response: Thank you for the suggestion, the two references have been added.

Reviewer #2: This manuscript presents a framework for mitigation of the interference in a heterogenous networks where both full-duplex (FD) and half-duplex (HD) relays are exploited. The novel contribution of this work is the consideration of three kinds of interference (inter-relay interference, residual self-interference, and relay-to-destination interference in a multi-cell environment, especially in a heterogenous network. The authors also solve a joint power allocation optimization problem for the relays and the sources.

In general, the technical contribution of this work is good and the analysis is presented in a good organization. However, I do have some comments to improve the quality of this manuscript:

1. Although the problem considered in this work is novel and comprehensive, the solutions to the considered problem, such as beamforming, zero-forcing, and singular value decomposition, have been well-studied in the previous works. The beamforming designs for the relays and the destinations are quite independent on each other. Therefore, they are not so difficult to derived.

Response: Beamforming have been used in different networks with different criteria. In this work, our joint analysis will not be possible without beamforming technique which simplifies the heterogeneous network analysis that equipped with MIMO antennas.

2. In the system model, the authors consider two adjacent cells, one of which uses half-duplex relay while the other uses full-duplex relay. This is not a typical situation. Why don't they consider two adjacent cells, both of which employ full-duplex relay?

Response: In cellular standards such as 5G and beyond and due to increased connectivity, in order to satisfy different quality-of-service constraints, ideally the network will comprise of half-duplex, full-deplex, IoT , drone devices, and so on. However, most recent research considered two adjacent cells employing either full–duplex or half-duplex relays. Hence, a heterogeneous network that comprised of multicell, HD and FD is much appreciated.

3. Regarding the joint power allocation for source and relay nodes, why is there the total power constraint? These are two independence devices, so each should has its own maximum available power.

Response: In order to optimise the proposed network as a whole system including the sources, relays and destinations, the total power constraint reflects the available power budget, one could easily replace “1” by a desired power budget.

4. The baseline schemes for comparison are the HD and FD schemes without interference or with interference but without interference cancellation. These comparisons are trivial. Please compare the proposed method with related works that the authors have introduced.

Response: The baseline schemes for comparison are the HD and FD schemes without interference or with interference have been derived from the literature, we simplify the names for the convenience, however, few references have added. 

5. In the Introduction section (page 3, lines 78 - 79), the authors claim that "by changing the level of self-interference, the FD ergodic capacity becomes similar to HD ergodic capacity". But then in the next sentence, they stated that "the ergodic capacity of FD achieves better performance than HD ergodic capacity". Please justify these sentences.

Response: The references have been clarified

6. In the comments on Figure 3, the authors claim that "With the increase in SNR, the performance of FD without interference cancellation decreases". This is not compatible with the results in Figure 3.

Response: This updated to “With the increase in SNR, the performance of FD without interference cancellation remains flat”

7. In page 11, line 192, what does C^{(M_{S_j}, M_{r_j}) x 1} means?

Response: The sentence is updated as follows “ The definitions of symbol vectors and channel matrices are shown in the following. \\mathbf{x}_{\\mathit{S}}\\in\\mathbb{C}^{min(M_{Si},M_{ri})\\times1}, and \\mathbf{\\hat{x}}_{\\mathit{S}}\\in\\mathbb{C}^{min(M_{Sj},M_{rj})\\times1} are the transmitted signals from the S node with dimension min\\,(M_{Si},M_{ri})\\times1 and R node with dimension min\\,(M_{Sj},M_{rj})\\times1, respectively”.

8. Page 8, line 148: "(x)^{+} denotes x or 0"  this is not correct mathemmatically.

Response: The sentence is updated as follows \\left[x\\right]^{+}\\triangleq max\\{x,0\\}.

9. There are many typos and grammar errors in the manuscript: - There are some duplication between the Introduction and the section on Related work, for example: "The main hurdle facing relay assisted ..." or "Small cells such as Femto and Pico ..." - Page 3, lines 84-86: "... relay systems Table 1". - Page 3, lines 101-102: "Though it achieves ... was not considered"  grammar error. - The first paragraph of the subsection "Summary of Contribution" is not clear in meaning. Also, there is a typo: "Table ??". - "moore-penrose pseudoinverse"  should be "Moore-Penrose ...".

Response: Thank you for pointing out the syntax errors. The duplicating sentences have been removed. Page 3, lines 84-86: "... relay systems Table 1" is changed to “....relay systems as shown in Table 1”. The first paragraph of the subsection "Summary of Contribution" is changed to “Contribution". "moore-penrose pseudoinverse"  is changed to "Moore-Penrose ...".

"Though it achieves ... was not considered" is updated to “...designed based on additional hardware. However, the system did not capture the network heterogeneity.

10. Table 1 is too long, but the readers still cannot see what is the difference between the proposed work and those works in Table 1.

Response: We have revised the final manuscript with the above suggestion and introduced the last row of the table to highlight our contribution.

---

## [Decision Letter · Decision Letter 1]

20 Mar 2023

PONE-D-22-34295R1Half-Duplex and Full-Duplex Interference Mitigation in Relays Assisted Heterogeneous NetworkPLOS ONE

Dear Dr. Madani Fadoul,

Thank you for submitting your manuscript to PLOS ONE. After careful consideration, we feel that it has merit but does not fully meet PLOS ONE’s publication criteria as it currently stands. Therefore, we invite you to submit a revised version of the manuscript that addresses the points raised during the review process.

We look forward to receiving your revised manuscript.

Kind regards,

Praveen Kumar Donta, Ph.D.

Academic Editor

PLOS ONE

Journal Requirements:

Reviewers' comments:

Reviewer's Responses to Questions

**Comments to the Author**

1. If the authors have adequately addressed your comments raised in a previous round of review and you feel that this manuscript is now acceptable for publication, you may indicate that here to bypass the “Comments to the Author” section, enter your conflict of interest statement in the “Confidential to Editor” section, and submit your "Accept" recommendation.

Reviewer #1: All comments have been addressed

Reviewer #2: (No Response)

2. Is the manuscript technically sound, and do the data support the conclusions?

Reviewer #1: Yes

Reviewer #2: Yes

3. Has the statistical analysis been performed appropriately and rigorously? 

Reviewer #1: Yes

Reviewer #2: Yes

4. Have the authors made all data underlying the findings in their manuscript fully available?

Reviewer #1: Yes

Reviewer #2: Yes

5. Is the manuscript presented in an intelligible fashion and written in standard English?

Reviewer #1: Yes

Reviewer #2: Yes

6. Review Comments to the Author

Reviewer #1: Authors have addressed all my comment's, not more comments. Authors have addressed all my comment's, not more comments. Authors have addressed all my comment's, not more comments.

Reviewer #2: In this revised manuscript, the authors have made some appropriate changes to improve the presentation of their work. However, some of technical concerns that I have commented in the previous review have not been responded seriously.

1. Regarding the comment on the assumption that the adjacent cell uses half-duplex transmission instead of full-duplex, please state clearly in the manuscript the practical situations that this work can be applied. This should be done to show a solid motivation for their work.

2. Regarding the question on the constrain on the total power constraint, it seems that the authors misunderstood my comment. I don't care what the values the authors set for the total power constraint (1 or any). My main concern is that any device should have their own power constraint, which does not depend on the power consumption of other devices. In your model, does the relay need to contact the base station to know about the maximum power that it (the relay) can transmit? Please clarify this setup by solid argument, not by references (because this should be specific to your model).

3. In response to the concern about the claim "by changing the level of self-interference, the FD ergodic capacity becomes like HD ergodic capacity", then "Finally, the ergodic capacity of FD achieves better performance than HD ergodic capacity", the authors referred to two previous references but actually, they were not related. So, please elaborate on those claims or rewrite the sentences to make it easier to follow.

In general, I expect more serious response from the authors on the technical points, not just 2-3 sentences and without any changes in the manuscript.

7. PLOS authors have the option to publish the peer review history of their article (what does this mean?). If published, this will include your full peer review and any attached files.

Reviewer #1: No

Reviewer #2: No

---

## [Author Response · Author response to Decision Letter 1]

10 Apr 2023

Manuscript ID PONE-D-22-34295R1

Firstly, we would like to thank the editor and the reviewers for their efforts in reviewing our manuscript. The comments are useful and constructive. Following the comments, we have made changes to improve the final manuscript. We also respond to individual concerns of the reviewers, where the comments are printed in italic, followed by the responses. In the revised manuscript, all changes are indicated by red colored fonts. The manuscript is edited in order to comply with length policy imposed by PLOS ONE Journal.

Reviewers' comments:

1. If the authors have adequately addressed your comments raised in a previous round of review and you feel that this manuscript is now acceptable for publication, you may indicate that here to bypass the “Comments to the Author” section, enter your conflict of interest statement in the “Confidential to Editor” section, and submit your "Accept" recommendation.

Reviewer #1: All comments have been addressed

Reviewer #2: (No Response)

2. Is the manuscript technically sound, and do the data support the conclusions? 

Reviewer #1: Yes

Reviewer #2: Yes

3. Has the statistical analysis been performed appropriately and rigorously?

Reviewer #1: Yes

Reviewer #2: Yes

4. Have the authors made all data underlying the findings in their manuscript fully available?

Reviewer #1: Yes

Reviewer #2: Yes

5. Is the manuscript presented in an intelligible fashion and written in standard English?

Reviewer #1: Yes

Reviewer #2: Yes

6. Review Comments to the Author

Reviewer #1: Authors have addressed all my comment's, not more comments. Authors have addressed all my comment's, not more comments. Authors have addressed all my comment's, not more comments.

Response: Thank you very much

Reviewer #2: In this revised manuscript, the authors have made some appropriate changes to improve the presentation of their work. However, some of technical concerns that I have commented in the previous review have not been responded seriously.

1. Regarding the comment on the assumption that the adjacent cell uses half-duplex transmission instead of full-duplex, please state clearly in the manuscript the practical situations that this work can be applied. This should be done to show a solid motivation for their work.

Response: We have revised the final manuscript with the above suggestion (lines 161 to 168).

2. Regarding the question on the constrain on the total power constraint, it seems that the authors misunderstood my comment. I don't care what the values the authors set for the total power constraint (1 or any). My main concern is that any device should have their own power constraint, which does not depend on the power consumption of other devices. In your model, does the relay need to contact the base station to know about the maximum power that it (the relay) can transmit? Please clarify this setup by solid argument, not by references (because this should be specific to your model).

Response: We have revised the final manuscript with the above clarifications (lines 327 to 336).

3. In response to the concern about the claim "by changing the level of self-interference, the FD ergodic capacity becomes like HD ergodic capacity", then "Finally, the ergodic capacity of FD achieves better performance than HD ergodic capacity", the authors referred to two previous references but actually, they were not related. So, please elaborate on those claims or rewrite the sentences to make it easier to follow.

In general, I expect more serious response from the authors on the technical points, not just 2-3 sentences and without any changes in the manuscript.

Response: We revised the final manuscript and highlighted the above suggestions (lines 77 to 83). Thank you for taking the time to improve the manuscript; we eagerly await your feedback.

---

## [Decision Letter · Decision Letter 2]

29 May 2023

Half-Duplex and Full-Duplex Interference Mitigation in Relays Assisted Heterogeneous Network

PONE-D-22-34295R2

Dear Dr. Madani Fadoul,

We’re pleased to inform you that your manuscript has been judged scientifically suitable for publication and will be formally accepted for publication once it meets all outstanding technical requirements.

Kind regards,

Praveen Kumar Donta, Ph.D.

Academic Editor

PLOS ONE

Additional Editor Comments (optional):

Reviewers' comments:

Reviewer's Responses to Questions

**Comments to the Author**

1. If the authors have adequately addressed your comments raised in a previous round of review and you feel that this manuscript is now acceptable for publication, you may indicate that here to bypass the “Comments to the Author” section, enter your conflict of interest statement in the “Confidential to Editor” section, and submit your "Accept" recommendation.

Reviewer #1: (No Response)

2. Is the manuscript technically sound, and do the data support the conclusions?

Reviewer #1: (No Response)

3. Has the statistical analysis been performed appropriately and rigorously? 

Reviewer #1: (No Response)

4. Have the authors made all data underlying the findings in their manuscript fully available?

Reviewer #1: (No Response)

5. Is the manuscript presented in an intelligible fashion and written in standard English?

Reviewer #1: (No Response)

6. Review Comments to the Author

Reviewer #1: (No Response)

7. PLOS authors have the option to publish the peer review history of their article (what does this mean?). If published, this will include your full peer review and any attached files.

Reviewer #1: No

---

## [Editor Report · Acceptance letter]

9 Jun 2023

PONE-D-22-34295R2 

Half-Duplex and Full-Duplex Interference Mitigation in Relays
Assisted Heterogeneous Network 

Dear Dr. Madani Fadoul:

I'm pleased to inform you that your manuscript has been deemed suitable for publication in PLOS ONE. Congratulations! Your manuscript is now with our production department. 

Kind regards, 

on behalf of

Dr. Praveen Kumar Donta 

Academic Editor

PLOS ONE